# Learning Macroscopic Dynamics from Partial Microscopic Observations

**Mengyi Chen[1], Qianxiao Li[1, 2]**
Department of Mathematics, National University of Singapore[1],
Institute for Functional Intelligent Materials, National University of Singapore[2]
`chenmengyi@u.nus.edu`, `qianxiao@nus.edu.sg`

## Abstract

Macroscopic observables of a system are of keen interest in real applications such as the design of novel materials. Current methods rely on microscopic trajectory simulations, where the forces on all microscopic coordinates need to be computed or measured. However, this can be computationally prohibitive for realistic systems. In this paper, we propose a method to learn macroscopic dynamics requiring only force computations on a subset of the microscopic coordinates. Our method relies on a sparsity assumption: the force on each microscopic coordinate relies only on a small number of other coordinates. The main idea of our approach is to map the training procedure on the macroscopic coordinates back to the microscopic coordinates, on which partial force computations can be used as stochastic estimation to update model parameters. We provide a theoretical justification of this under suitable conditions. We demonstrate the accuracy, force computation efficiency, and robustness of our method on learning macroscopic closure models from a variety of microscopic systems, including those modeled by partial differential equations or molecular dynamics simulations. Our code is available at `https://github.com/MLDS-NUS/Learn-Partial.git`.

## 1 Introduction

Macroscopic properties, including thestructural and dynamical properties, provide a way to describe and understand the collective behaviors of complex systems. In a wide range of real applications, researchers focus mainly on the macroscopic properties of a system, *e.g.*, the viscosity and ionic diffusivity of liquid electrolytes for Li-ion batteries (Dajnowicz et al., 2022). Macroscopic observables usually depend on the whole microscopic system, *e.g.*, the calculation of mean squared displacement requires all microscopic coordinates during the simulation. With growing simulation and experimental data, data-driven learning of macroscopic properties from microscopic observations has become an active area of research (Zhang et al., 2018; Wang et al., 2019; Husic et al., 2020; Lee et al., 2020; Fu et al., 2023; Chen et al., 2024).

Accurate calculation of macroscopic properties requires large-scale microscopic simulation. However, accurate force computations on all microscopic coordinates for large systems are extremely expensive (Jia et al., 2020; Musaelian et al., 2023). For example, in *ab initio* molecular simulations, accurate forces need to be calculated from density functional theory (DFT). The computational cost of DFT limits its application to relatively small systems, typically ranging from a few hundred atoms to several thousand atoms, depending on the level of accuracy and computation resources (Hafner et al., 2006; Luo et al., 2020). This poses a dilemma: Accurate macroscopic properties are obtained from large-scale microscopic simulation, but the computation of forces on all the microscopic coordinates is extremely challenging.

38th Conference on Neural Information Processing Systems (NeurIPS 2024).

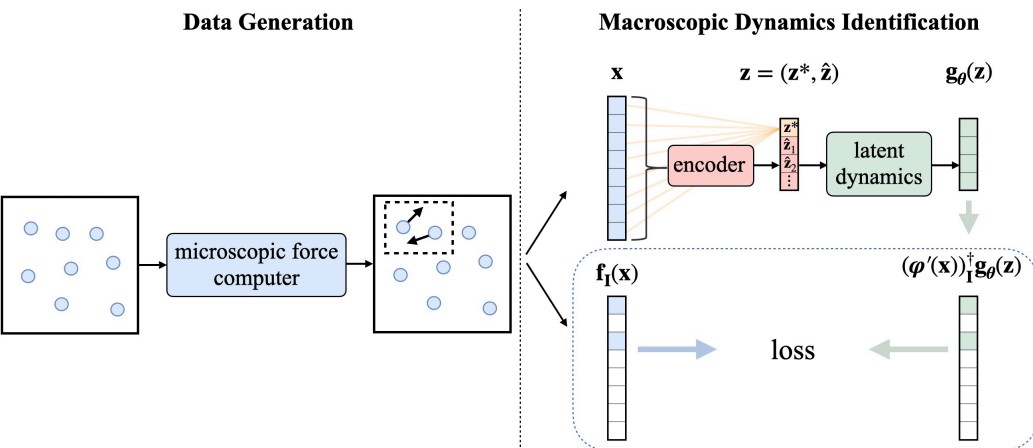

Figure 1: Overview of our method. *Left*. Data generation workflow. For each configuration $\mathbf{x}$, forces on a subset of all the microscopic coordinates are calculated by the microscopic force calculator. *Right*. Macroscopic dynamics identification. The macroscopic dynamics is mapped to the microscopic space first, then compared with the forces on a subset of the microscopic coordinates.

To solve the dilemma, the corresponding question is: Can we still obtain accurate macroscopic observables even though only access to forces on a subset of the microscopic coordinates? In this work, we develop a method to learn the dynamics of the macroscopic observables directly, while only forces on a subset of the microscopic coordinates are needed. Efficient partial computation of microscopic forces relies on the sparsity assumption, where the computation cost of forces on a subset of microscopic coordinates does not scale with the microscopic system size. To learn the dynamics of the macroscopic observables, we first map the macroscopic dynamics back to the microscopic space, then compare it with the partial microscopic forces. Our key idea is summarized in Fig. 1.

Our main contributions are as follows:

- We develop a novel method that can learn the macroscopic dynamics from partial computation of the microscopic forces. Our method can significantly reduce the computational cost for force computations.

- We theoretically justify that forces on a subset of the microscopic coordinates can be used as stochastic estimation to update latent model parameters, even if the macroscopic observables depend on all the microscopic coordinates.

- We empirically validate the accuracy, force computation efficiency, and robustness of our method through a variety of microscopic dynamics and latent model structures.

## 2 Related Work

**Learning from Partial Observations** Several works have sought to learn dynamics from partially observed state $\hat{\mathbf{x}}$ utilizing machine learning (Ruelle & Takens, 1971; Sauer et al., 1991; Takens, 2006; Ayed et al., 2019; Ouala et al., 2020; Huang et al., 2020; Schlaginhaufen et al., 2021; Lu et al., 2022; Stepaniants et al., 2023). For training, these methods reconstruct the unobserved state $\tilde{\mathbf{x}}$ first and model the dynamics of $\mathbf{x} = (\hat{\mathbf{x}}, \tilde{\mathbf{x}})$. Our work assumes full state $\mathbf{x}$ but partial forces $\mathbf{f}$. Furthermore, we do not model the dynamics on state $\mathbf{x}$ directly, but rather on the latent space because the dimension of $\mathbf{x}$ would be extremely high for large systems.

**Reduced Order Models** By modeling the dynamics in the latent space and then recovering the full states from them, reduced order models (ROMs) substitute expensive full order simulation with cheaper reduced order simulation (Schilders et al., 2008; Fresca et al., 2020; Lee & Carlberg, 2020; Hernandez et al., 2021; Fries et al., 2022).

Our method can be thought to fall in the range of closure modeling. Unlike ROMs, we aim to model the dynamics of some given macroscopic observables directly and are not interested in recovering the microscopic states from the latent states.

**Equation-free Framework** The equation-free framework (EFF) has sought to simulate the macroscopic dynamics efficiently (Kevrekidis et al., 2003; Samaey et al., 2006; Liu et al., 2015). The EFF is usually applied to partial differential equation (PDE) systems, and the macroscopic observables are chosen to be the solution of the PDE at the coarse spatial grid. In EFF, the macroscopic observables depend locally on the microscopic coordinates, allowing the macroscopic dynamics to be directly estimated from the microscopic simulations performed in small spatial domains. In contrast, the macroscopic observables may depend globally on the microscopic coordinates in our method, and the macroscopic dynamics may not be easily estimated from microscopic simulations performed in small spatial domains.

Another difference is that our method explicitly learns the macroscopic dynamics, while EFF can bypass explicit derivation of macroscopic evolution law by coupling microscale and macroscale dynamics. During simulation, EFF still requires microscopic simulation to be performed in small spatial domains and for short times, but our method can enable fast macroscopic simulation without requiring any microscopic simulation However, for systems where the macroscopic evolution equations conceptually exist but are not available in closed form, EFF can efficiently handle such cases, but the learned dynamics in our method may involve approximation or statistical errors that are often challenging to estimate.

## 3 Problem Setup

We consider a microscopic system consisting of $n$ particles. Let the state of the microscopic system be $\mathbf{x} = (\mathbf{x}_1, \cdots, \mathbf{x}_n) \in \mathbb{R}^N, \mathbf{x}_i \in \mathbb{R}^m, N = mn$, where $\mathbf{x}_i \in \mathbb{R}^m$ is some physical quantity associated with the $i$-th particle, such as the position and velocity. Assume the dynamics of the microscopic system can be characterized by an ordinary differential equation(ODE):

$$\frac{\mathrm{d}\mathbf{x}(t)}{\mathrm{d}t} = \mathbf{f}(\mathbf{x}(t)) \tag{1}$$

where $\mathbf{f}(\mathbf{x}) = (\mathbf{f}_1(\mathbf{x}), \cdots, \mathbf{f}_n(\mathbf{x})) \in \mathbb{R}^N$. We will call $\mathbf{x}_i$ the *microscopic coordinate* of the $i$-th particle and $\mathbf{f}_i$ the *force* acting on the microscopic coordinate $\mathbf{x}_i$. In many real applications, we are interested in the dynamics of some macroscopic observables $\mathbf{z}^* = \boldsymbol{\varphi}^*(\mathbf{x})$. Here $\boldsymbol{\varphi}^*$ is given beforehand and describes the functional dependence of $\mathbf{z}^*$ on $\mathbf{x}$. For example, $\mathbf{z}^*$ can be chosen to be the instantaneous temperature or mean squared displacement in a Lennard-Jones system.

The goal is to learn the dynamics of $\mathbf{z}^*$ from microscopic simulation data. Existing methods that try to learn the macroscopic or latent dynamics require microscopic trajectories or forces on all the microscopic coordinates for training (Champion et al., 2019; Fries et al., 2022; Fu et al., 2023; Chen et al., 2024). The problem is: When the microscopic system size $N$ is very large such that the force computations on all the microscopic coordinates are impossible, these methods are no longer applicable. Instead, our method aims to learn from partial computation of microscopic forces.

Consider we are given a microscopic force calculator $\mathcal{S}$ for computation of partial microscopic forces. Let the microscopic coordinate $\mathbf{x}$ be sampled from a distribution $\mathcal{D}$. For each $\mathbf{x}$, the microscopic force calculator $\mathcal{S}$ will first sample an $n$ dimensional random variable $\mathbf{I}(\mathbf{x}) = (\mathbf{I}_1(\mathbf{x}), \cdots, \mathbf{I}_n(\mathbf{x})) \sim \mathcal{P}_{\mathbf{x}} \in \{0, 1\}^n$ according to a certain strategy. Next $\mathcal{S}$ will calculate the corresponding *partial forces* $\mathbf{f}_{\mathbf{I}(\mathbf{x})} := (\mathbf{f}_{\mathbf{I}_1(\mathbf{x})}, \cdots, \mathbf{f}_{\mathbf{I}_n(\mathbf{x})})$. For notation simplicity sometimes we will simply write $\mathbf{f}_{\mathbf{I}(\mathbf{x})}$ as $\mathbf{f}_{\mathbf{I}}$. $\mathbf{I}_i(\mathbf{x})$ indicate whether partial $i$ is chosen to calculate the force or not. If particle $i$ is chosen, $\mathbf{I}_i(\mathbf{x}) = 1, \mathbf{f}_{\mathbf{I}_i(\mathbf{x})} = \mathbf{f}_i$, otherwise $\mathbf{I}_i(\mathbf{x}) = 0, \mathbf{f}_{\mathbf{I}_i(\mathbf{x})} = \mathbf{0}$. We require the sampling strategy $\mathcal{P}_{\mathbf{x}}$ to satisfy:

1. For each $\mathbf{I}(\mathbf{x}) \sim \mathcal{P}_{\mathbf{x}}$, exactly $n \cdot p$ items are equal to 1 and the rest are 0.
2. Each particle can be chosen with equal probability p, *i.e.* $\mathbb{P}(\mathbf{I}_i(\mathbf{x}) = 1) = p, \mathbb{P}(\mathbf{I}_i(\mathbf{x}) = 0) = 1 - p, i = 1, \cdots, n$.

This means that the microscopic force calculator $\mathcal{S}$ can calculate forces on $n \cdot p$ microscopic coordinates, and $0 < p < 1$ limits the computation capacity of the microscopic force calculator $\mathcal{S}$.

The above requirement is consistent with real applications since it is difficult to calculate all the microscopic forces due to computational cost. Furthermore, for efficient calculation of the partial microscopic forces, we will assume $\mathbf{f}$ satisfies the following sparsity assumption:

**Assumption**: For a given error tolerance $\epsilon > 0$, there exists a constant $M \ll n$, such that for any $\mathbf{x} \sim \mathcal{D}$ and $i \in \{1, \cdots, n\}$, we can always find an index set $J(\mathbf{x}_i) \subset \{i = 1, \cdots, n\}, |J(\mathbf{x}_i)| < M$ which satisfies:

$$||\mathbf{f}_i(\mathbf{x}_1, \cdots, \mathbf{x}_n) - \tilde{\mathbf{f}}_i(\{\mathbf{x}_i\}_{i \in J(\mathbf{x}_i)})||_2 < \epsilon \tag{2}$$

Intuitively, the assumption implies that the computational cost of force $\mathbf{f}_i$ is independent of the microscopic system dimension $N$. Thus our microscopic force calculator $\mathcal{S}$ can compute partial forces in an efficient way. This assumption is prevalent in real-world applications. To better illustrate this, we give two examples here. The first example is about molecular dynamics. In molecular dynamics, each $\mathbf{x}_i$ represents the position $\mathbf{r}_i$ and velocities $\mathbf{v}_i$ of the $i$-th atom, *i.e.*, $\mathbf{x}_i = (\mathbf{r}_i, \mathbf{v}_i) \in \mathbb{R}^6$. Then Eq. (1) becomes the Newton's law of motion:

$$\frac{\mathrm{d}\mathbf{r}_i}{\mathrm{d}t} = \mathbf{v}_i \tag{3}$$

$$\frac{\mathrm{d}\mathbf{v}_i}{\mathrm{d}t} = \frac{1}{m_i}\mathbf{F}_i(\mathbf{r}_1, \cdots, \mathbf{r}_n), \tag{4}$$

It is common to limit the range of pairwise interactions to a cutoff distance (Allen et al., 2004; Zhou & Liu, 2022; Vollmayr-Lee, 2020). To calculate the force on an atom, we only need to consider its interaction with other atoms that are within the cutoff. The second example is about systems modeled by partial differential equation (PDE). We consider a time-dependent PDE and apply finite difference scheme to discretize the spatial derivatives. Then, the resulting semi-discretized equation takes the form of Eq. (1), and each $\mathbf{x}_i$ is the value at the $i$-th grid. $\mathbf{f}_i$ only depends on those grids that are used for finite difference approximation of the spatial derivatives.

Let the training data generated by the microscopic force calculator $\mathcal{S}$ be $\{\mathbf{x}^i, \mathbf{f}_{\mathbf{I}(\mathbf{x}^i)}\}_{i=1,\cdots,K}$. The data generation procedure is provided in Algorithm 1. We will introduce in the next section how we can learn the macroscopic dynamics from the training data with partial forces.

## 4 Method

Existing works for macroscopic dynamics identification consist of two parts: dimension reduction and macroscopic dynamics identification (Fu et al., 2023; Chen et al., 2024). We will follow these two parts. We start with most standard parts of closure modeling with an autoencoder, next, we turn to the main difficulty of macroscopic dynamics identification from partial forces.

### 4.1 Autoencoder for Closure Modeling

We will use an autoencoder to find the closure $\hat{\mathbf{z}} = \hat{\varphi}(\mathbf{x})$ to $\mathbf{z}^* = \varphi^*(\mathbf{x})$ such that $\mathbf{z} = (\mathbf{z}^*, \hat{\mathbf{z}})$ forms a closed system. Here we define $\mathbf{z}$ as forming a closed system if its dynamics $\dot{\mathbf{z}}$ depends only on $\mathbf{z}$, not any external variables. Note that in $\mathbf{z}^* = \varphi^*(\mathbf{x})$, $\varphi^*$ is determined beforehand and contains no trainable parameters. This ensures $\mathbf{z}^*$ represents the desired macroscopic observables and remains unchanged during the training of the autoencoder.

Denote the encoder by $\varphi = (\varphi^*, \hat{\varphi})$ and the decoder by $\psi$, we will minimize the following reconstruction loss:

$$\mathcal{L}_{\mathrm{rec}} = \frac{1}{K}\sum_{i=1}^{K}||\mathbf{x}^i - \psi \circ \varphi(\mathbf{x}^i)||_2^2 \tag{5}$$

We also want $\varphi'(\mathbf{x})\varphi'(\mathbf{x})^T$ to be well-conditioned (see Section 4.2), then we impose constraints on the condition number of $\varphi'(\mathbf{x})\varphi'(\mathbf{x})^T$:

$$\mathcal{L}_{\mathrm{cond}} = \frac{1}{K}\sum_{i=1}^{K}||\kappa(\varphi'(\mathbf{x}^i)\varphi'(\mathbf{x}^i)^T) - 1||_2^2 \tag{6}$$

By enforcing $\varphi'(\mathbf{x}^i)\varphi'(\mathbf{x}^i)^T$ to be well-conditioned, we are also enforcing $\varphi'(\mathbf{x}^i) \in \mathbb{R}^{d \times N}, d \ll N$ to have full row rank, which will be used later. The overall loss to train the autoencoder is :

$$\mathcal{L}_{\mathrm{AE}} = \mathcal{L}_{\mathrm{rec}} + \lambda_{\mathrm{cond}}\mathcal{L}_{\mathrm{cond}}, \tag{7}$$

| **Algorithm 1** Data generation. | **Algorithm 2** Training procedure. |
|---|---|
| **Require:** | **Require:** |
| $\quad\mathcal{D}$:configuration distribution | $\quad\{\mathbf{x}^i,\mathbf{f}_{\mathbf{x}^i}(\mathbf{x}^i)\}_{i=1,\cdots,K}$: data |
| $\quad\mathcal{S}$: microscopic force calculator | $\quad B$: minibatch size |
| $\quad K$: training data size | $\quad\boldsymbol{\theta}_0$: model parameter |
| $\quad\mathcal{P}$: partial index sampling strategy | $\quad$opt: optimizer |
| 1: **for** $i=1$ **to** $K$ **do** | 1: **while** stopping criterion is not met **do** |
| 2: $\quad\mathbf{x}^i\sim\mathcal{D}$ | 2: $\quad$sample $J\subset\{1,\cdots,K\},|J|=B$ |
| 3: $\quad\mathbf{I}(\mathbf{x}^i)\sim\mathcal{P}_{\mathbf{x}^i}$ | 3: $\quad$Calculate $\mathcal{L}_{\mathbf{x},p}$ in Eq. (12) with $\{\mathbf{x}^i,\mathbf{f}_{\mathbf{x}^i}(\mathbf{x}^i)\}_{i\in J}$ |
| 4: $\quad$calculate $\mathbf{f}_{\mathbf{I}(\mathbf{x}^i)}(\mathbf{x}^i)$ using $\mathcal{S}$ | 4: $\quad\boldsymbol{\theta}_{t+1}\leftarrow\text{opt}(\boldsymbol{\theta}_t,\nabla\mathcal{L}_{\mathbf{x},p})$ |
| $\quad$**return** $\{\mathbf{x}^i,\mathbf{f}_{\mathbf{I}(\mathbf{x}^i)}(\mathbf{x}^i)\}_{i=1,\cdots,K}$ | $\quad$**return** model parameter |

$\lambda_{\text{cond}}$ is a hyperparameter to adjust the ratio of $\mathcal{L}_{\text{cond}}$ and is chosen to be quite small in our experiments, *e.g.*, $10^{-5}$ or $10^{-6}$. The aim of training the decoder $\psi$ is to help the discovery of the closure variables. The decoder $\psi$ will not be used for further macroscopic dynamics identification. To facilitate comparison between models tainted with all and partial forces, we will train the autoencoder first and freeze it for macroscopic dynamics identification.

## 4.2 Macroscopic Dynamics Identification

We now address the difficulty of learning from data with partial microscopic forces. Substitute $\mathbf{z}=\boldsymbol{\varphi}(\mathbf{x})$ into equation Eq. (1) and make use of chain rule, we get the dynamics of $\mathbf{z}$:

$$\frac{d\mathbf{z}}{dt}=\boldsymbol{\varphi}'(\mathbf{x})\mathbf{f}(\mathbf{x}),\quad\mathbf{z}(0)=\boldsymbol{\varphi}(\mathbf{x}_0)\tag{8}$$

here we use $\boldsymbol{\varphi}'(\mathbf{x})$ to denote the Jacobian of $\nabla_{\mathbf{x}}\boldsymbol{\varphi}(\mathbf{x})$ for notation simplicity. If the dynamics of $\mathbf{z}$ is closed, the right-hand side of Eq. (8) will only depend on $\mathbf{z}$, and we parametrize it using a neural network $\mathbf{g}_{\boldsymbol{\theta}}(\mathbf{z})\approx\boldsymbol{\varphi}'(\mathbf{x})\mathbf{f}(\mathbf{x})$. Since we are only interested in macroscopic dynamics identification, the loss would be naturally defined on the macroscopic coordinates:

$$\mathcal{L}_{\mathbf{z}}(\boldsymbol{\theta})=\frac{1}{K}\sum_{i=1}^K||\boldsymbol{\varphi}'(\mathbf{x}^i)\mathbf{f}(\mathbf{x}^i)-\mathbf{g}_{\boldsymbol{\theta}}(\mathbf{z}^i)||^2\tag{9}$$

Eq. (9) is used commonly in existing work (Champion et al., 2019; Fries et al., 2022; Bakarji et al., 2022; Park et al., 2024).

The main difficulty of $\mathcal{L}_{\mathbf{z}}$ is that it includes the matrix-vector product $\boldsymbol{\varphi}'(\mathbf{x})\mathbf{f}(\mathbf{x})$. Note that the $i$-th entry of $\boldsymbol{\varphi}'(\mathbf{x})\mathbf{f}(\mathbf{x})$ can be written as $\sum_{j=1}^n\varphi'_{ij}(\mathbf{x})\mathbf{f}_j(\mathbf{x})$, and it is difficult to find an unbiased estimation of the $i$-th entry using a subset of $\{\mathbf{f}_j(\mathbf{x})\}_{j=1,\cdots,N}$. Thus the accurate calculation of $\mathcal{L}_{\mathbf{z}}$ requires the forces $\{\mathbf{f}_j(\mathbf{x})\}_{j=1,\cdots,N}$ on all the microscopic coordinates.

The main idea of our method is to map the loss on the macroscopic coordinates back to the microscopic coordinates:

$$\mathcal{L}_{\mathbf{x}}(\boldsymbol{\theta})=\frac{1}{K}\sum_{i=1}^K||\mathbf{f}(\mathbf{x}^i)-(\boldsymbol{\varphi}'(\mathbf{x}^i))^{\dagger}\mathbf{g}_{\boldsymbol{\theta}}(\mathbf{z}^i)||^2\tag{10}$$

where $(\boldsymbol{\varphi}'(\mathbf{x}^i))^{\dagger}\in\mathbb{R}^{N\times d}$ is the Moore-Penrose inverse. Since $\boldsymbol{\varphi}'(\mathbf{x}^i)$ is of full row rank, $(\boldsymbol{\varphi}'(\mathbf{x}^i))^{\dagger}$ is in fact the right inverse of $\boldsymbol{\varphi}'(\mathbf{x}^i)$, *i.e.*, $\boldsymbol{\varphi}'(\mathbf{x}^i)(\boldsymbol{\varphi}'(\mathbf{x}^i))^{\dagger}$ is an identity matrix. Below we will show our main theoretical result:

**Theorem 1.** *Assume for any* $\mathbf{x}\sim\mathcal{D}$, *the eigenvalues of* $\boldsymbol{\varphi}'(\mathbf{x})\boldsymbol{\varphi}'(\mathbf{x})^T$ *are lower bounded by* $b_1$ *and upper bounded by* $b_2$, $0<b_1\leqslant b_2$. *Then:*

$$b_1(\mathcal{L}_{\mathbf{x}}(\boldsymbol{\theta})+C)\leqslant\mathcal{L}_{\mathbf{z}}(\boldsymbol{\theta})\leqslant b_2(\mathcal{L}_{\mathbf{x}}(\boldsymbol{\theta})+C)\tag{11}$$

*here C does not depend on* $\boldsymbol{\theta}$ *hence does not affect the optimization.*

The proof relies on singular value decomposition of $\boldsymbol{\varphi}'(\mathbf{x})$ and we provide the full proof in Appendix A.1. Theorem 1 states that by minimizing $\mathcal{L}_{\mathbf{x}}(\boldsymbol{\theta})$, we are actually narrowing the range of $\mathcal{L}_{\mathbf{z}}(\boldsymbol{\theta})$. Hence we want $b_1$ and $b_2$ to be as close as possible, this is the reason why we constrain the

condition number of $\varphi'(\mathbf{x})\varphi'(\mathbf{x})^T$ in Eq. (6). In the very special case where $b_1 = b_2$, minimizing $\mathcal{L}_{\mathbf{x}}(\boldsymbol{\theta})$ is just equivalent to minimizing $\mathcal{L}_{\mathbf{z}}(\boldsymbol{\theta})$.

Note that in loss $\mathcal{L}_{\mathbf{x}}$, the term $||\mathbf{f}(\mathbf{x}) - (\varphi'(\mathbf{x}))^\dagger \mathbf{g}_\theta(\mathbf{z})||$ can be rewritten as $\sum_{j=1}^n ||\mathbf{f}_j(\mathbf{x}) - (\varphi'(\mathbf{x}))_j^\dagger \mathbf{g}_\theta(\mathbf{z})||$, and $\frac{1}{p}\sum_{j\in\mathbf{I}(\mathbf{x})} ||\mathbf{f}_j(\mathbf{x}) - (\varphi'(\mathbf{x}))_j^\dagger \mathbf{g}_\theta(\mathbf{z})||$ can be regarded as its unbiased stochastic estimation. Then, we can introduce our loss defined for partial forces:

$$\mathcal{L}_{\mathbf{x},p}(\boldsymbol{\theta}) = \frac{1}{pK}\sum_{i=1}^K ||\mathbf{f}_{\mathbf{I}(\mathbf{x}^i)}(\mathbf{x}^i) - (\varphi'(\mathbf{x}^i))_{\mathbf{I}(\mathbf{x}^i)}^\dagger \mathbf{g}_\theta(\mathbf{z}^i)||_2^2 \tag{12}$$

Here a constant $1/p$ is multiplied to $\mathcal{L}_{\mathbf{x},p}$ to guarantee:

$$\mathbb{E}_{\mathbf{x}^1,\cdots,\mathbf{x}^K}\mathbb{E}_{\mathbf{I}(\mathbf{x}^1),\cdots,\mathbf{I}(\mathbf{x}^K)}\mathcal{L}_{\mathbf{x},p}(\boldsymbol{\theta}) = \mathbb{E}_{\mathbf{x}^1,\cdots,\mathbf{x}^K}\mathcal{L}_{\mathbf{x}}(\boldsymbol{\theta}) \tag{13}$$

We provide the full proof of Eq. (13) in Appendix A.2. By training with the model with $\mathcal{L}_{\mathbf{x},p}$, we can use data with partial forces as stochastic estimation to update model parameters. The full training procedure is provided in Algorithm 2.

Note that in Algorithm 1 during the data generation, $\mathbf{I}(x^i)$ is also sampled from its distribution. Thus, $\mathcal{L}_{\mathbf{x},p}$ is deterministic once the samples $\{\mathbf{x}^i, \mathbf{f}_{\mathbf{I}(\mathbf{x}^i)}(\mathbf{x}^i)\}_{i=1,\cdots,K}$ are generated. In the limit, the estimation is unbiased:

**Theorem 2** (informal). *Let* $\tilde{\mathcal{L}}_{\mathbf{x}}(\boldsymbol{\theta}) = \mathbb{E}\mathcal{L}_{\mathbf{x}}(\boldsymbol{\theta}), \boldsymbol{\theta}^* \in \arg\min_{\boldsymbol{\theta}} \tilde{\mathcal{L}}_{\mathbf{x}}(\boldsymbol{\theta}), \boldsymbol{\theta}_{K,p} \in \arg\min_{\boldsymbol{\theta}} \mathcal{L}_{\mathbf{x},p}(\boldsymbol{\theta}),$ *then under certain conditions:*

$$\tilde{\mathcal{L}}_{\mathbf{x}}(\boldsymbol{\theta}_{K,p}) - \tilde{\mathcal{L}}_{\mathbf{x}}(\boldsymbol{\theta}^*) \xrightarrow{a.s.} 0 \tag{14}$$

The proof utilized Rademacher complexity and a crucial assumption used in the proof is the uniform boundedness of $\mathcal{L}_{\mathbf{x},p}$. The formal version of Theorem 2 and the complete proof is provided in Appendix A.3. Theorem 2 theoretically justifies the expected risk $\tilde{\mathcal{L}}_{\mathbf{x}}(\boldsymbol{\theta}_{K,p})$ at the optimal parameter found by $\mathcal{L}_{\mathbf{x},p}$, converges to the optimal expected risk $\tilde{\mathcal{L}}_{\mathbf{x}}(\boldsymbol{\theta}^*)$ as $K$ goes to infinity.

## 5 Experiments

In this section, we experimentally validate the accuracy, force computation efficiency, and robustness through a variety of microscopic dynamics.

### 5.1 Force Computation Efficiency

We first consider a one-dimensional spatiotemporal Predator-Prey system, mainly to validate the correctness and force computations efficiency of our method.

**Predator-Prey System**  The simplified form of the Predator-Prey system with diffusion (Murray, 2003) is

$$\begin{aligned} \frac{\partial u}{\partial t} &= u(1 - u - v) + D\frac{\partial^2 u}{\partial x^2} \\ \frac{\partial v}{\partial t} &= av(u - b) + \frac{\partial^2 v}{\partial x^2}, \quad x \in \Omega = [0,1], \quad t \in [0,\infty) \end{aligned} \tag{15}$$

where $u, v$ denote the dimensionless populations of the prey and predator respectively, $a, b, D$ are three parameters. The complex dynamics of Predator-Prey interaction, including the pursuit of the predator and the evasion of the prey in ecosystems, can be described by Eq. (15).

We discretize the spatial domain of Eq. (15) into 50 uniform grids with $x_i = (i - \frac{1}{2})\Delta x, \Delta x = 0.02, 1 \leqslant i \leqslant 50$. Let $\mathbf{u}(t) = (u(x_1, t), \cdots, u(x_{50}, t)), \mathbf{v}(t) = (v(x_1, t), \cdots, v(x_{50}, t))$, then $(\mathbf{u}(t), \mathbf{v}(t)) \in \mathbb{R}^{100}$ are treated as the microscopic states. After approximating the spatial derivatives in Eq. (15) with the finite difference method, we consider the semi-discrete equation which is an $N = 100$ dimensional ODE as our microscopic evolution law (see Appendix B.1).

We choose the macroscopic observable of interest to be $\mathbf{z}^* = (\bar{u}, \bar{v})$, the spatial average of the predator and the prey's population:

$$\bar{u} = \frac{1}{50}\sum_{i=1}^{50} u(x_i, t), \quad \bar{v} = \frac{1}{50}\sum_{i=1}^{50} v(x_i, t) \tag{16}$$

We find another 2 closure variables using the autoencoder, then the total dimension of the latent space $\mathbf{z}$ is 4. We choose $\mathcal{D}$ to be the trajectory distribution of the state $\mathbf{x}$. For the data generation with partial forces, given $\mathbf{x}$ we randomly choose forces on $100 \cdot p$ microscopic coordinate. For example, if $p = 1/5$, then for each configuration, forces on 20 coordinates are calculated for training.

**Results** Fig. 2 shows the results on the test dataset which consists of 100 trajectories (for more detials, see Appendix Table 2). *For models trained with partial microscopic forces, we report the equivalent number of training data with full forces throughout the paper.* For example, for a model trained with $\mathcal{L}_{\mathbf{x},p}(p = 1/5)$ on $3 \times 10^3$ data, we will report the number of training data to be $3 \times 10^3 \times 0.2 = 600$. The test error is defined to be the mean relative error of the macroscopic observables between the ground truth and the predicted trajectories as in Appendix Eq. (43).

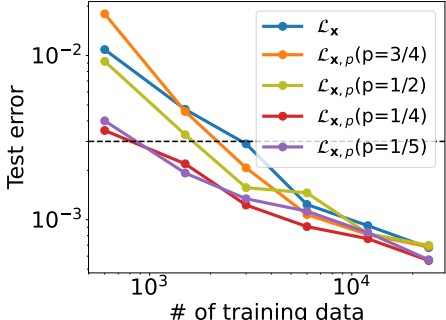

Figure 2: Mean relative error on the test dataset of the Predator-Prey system. The black dashed line represents test error = $3 \times 10^{-3}$.

First, we observe that the mean relative error of all the models is around $10^{-4}$ when the number of training data is large enough. This tells us that training with $\mathcal{L}_{\mathbf{x},p}$ is correct and accurate, which is consistent with Theorem 2. The predicted trajectories fit quite well with the ground truth trajectories (see Appendix Fig. 6 and Fig. 7).

We can conclude from Fig. 2 that, under the same number of training data, $\mathcal{L}_{\mathbf{x},p}$ with smaller $p$ $(1/4, 1/5)$ performs better. Similarly, to achieve the same performance, $\mathcal{L}_{\mathbf{x},p}$ with smaller $p$ requires less training data. We set the error tolerance to be $e_{\text{tol}} = 3 \times 10^{-3}$ and investigate how much training data is required to reach the error tolerance. In Fig. 2 the $x$-coordinate of the intersection point between the black dashed line and the other curves indicates the minimum data size required If we arrange each model according to their minimal required training data, then $\mathcal{L}_{\mathbf{x},p}(p = 1/5) \approx \mathcal{L}_{\mathbf{x},p}(p = 1/4) < \mathcal{L}_{\mathbf{x},p}(p = 1/2) < \mathcal{L}_{\mathbf{x},p}(p = 3/4) < \mathcal{L}_{\mathbf{x}}$. Model trained with $\mathcal{L}_{\mathbf{x},p}(p = 1/4, 1/5)$ requires less data to reach $e_{\text{tol}}$, or equivalently, less force computations. This validates the *force computation efficiency* of our method. One explanation could be that there are many redundant information in the forces acting on all the microscopic coordinates. By using partial microscopic forces, $\mathcal{L}_{\mathbf{x},p}$ can explore more configurations $\mathbf{x}$ given the same size of training data, thus can make use of more useful information. Another observation from Fig. 2 is that as the training data size increases, the gap between models trained with different $p$ narrows down. This is because as more data are provided, these data can contain almost all the information of the Predator-Prey system, thus more information will not lead to significant improvement.

## 5.2 Robustness to Different Latent Structures

Having validated the correctness and force computation efficiency of our method, we are ready to apply our method to a variety of latent structures. We will tackle the Lennard-Jones system in this subsection, and validate the robustness of our method to different latent model structures. Three latent model structures are considered: MLP, OnsagerNet (Yu et al., 2021), GFINNs (Zhang et al., 2022). Both the OnsagerNet and GFINNs endow the latent dynamical model with certain thermodynamic structure to ensure stability and interpretability. The specific implementations of these two models are slightly different.

**Lennard-Jones System** The Lennard-Jones system is widely used in molecular simulation to study phase transition, crystallization and macroscopic properties of a system (Hansen & Verlet, 1969; Bengtzelius, 1986; Lin et al., 2003; Luo et al., 2004). The Lennard-Jones potential describes the interaction between two atoms $i$ and $j$ through the potential of the following form:

$$V_{ij}(r) = \begin{cases} 4\epsilon_{ij}[(\sigma_{ij}/r)^{12} - (\sigma_{ij}/r)^6] & \text{if } r \leqslant r_{\text{cut}}, \\ 0 & \text{if } r > r_{\text{cut}}. \end{cases} \tag{17}$$

Table 1: Summary of the results on each system. Results of the Predator-Prey and Lennard-Jones (small) system are taken from Section 5.1, Section 5.2. For each system, $\mathcal{L}_{\mathbf{z}}$ and $\mathcal{L}_{\mathbf{x},p}$ are trained with the same size of data.

| | Micro dim $N$ | Observables | Latent dim $d$ | Partial labels $p$ | $\mathcal{L}_{\mathbf{z}}$ | $\mathcal{L}_{\mathbf{x},p}$ |
|---|---|---|---|---|---|---|
| Predator-Prey system | 100 | $\bar{u}, \bar{v}$ | 4 | 1/5 | $3.19_{\pm0.60}\times10^{-3}$ | $\mathbf{1.34}_{\pm0.16}\times\mathbf{10^{-3}}$ |
| Allen-Cahn system | 40000 | free energy $\mathcal{E}(v)$ | 16 | 1/25 | $6.93_{\pm2.80}\times10^{-3}$ | $\mathbf{3.98}_{\pm1.58}\times\mathbf{10^{-3}}$ |
| Lennard-Jones system (small) | 4800 | temperature $T$ | 32 | 1/16 | $4.45_{\pm2.03}\times10^{-3}$ | $\mathbf{1.17}_{\pm0.18}\times\mathbf{10^{-3}}$ |
| Lennard-Jones system (large) | 307200 | temperature $T$ | 32 | 1/1024 | - | $\mathbf{4.96}_{\pm0.56}\times\mathbf{10^{-3}}$ |

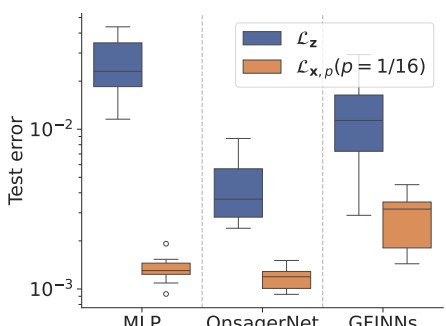

All the results in this experiment will be shown in reduced Lennard-Jones units. We consider a three-dimensional Lennard-Jones fluid with $N_{\text{atoms}} = 800$ atoms of the same type. The microscopic state consists of the positions and velocities of the 800 atoms, thus the microscopic dimension is $N = 4800$. We simulate the Lennard-Jones system under NVE ensemble using LAMMPS (Thompson et al., 2022).

We choose the instantaneous temperature ($T$) as our macroscopic observables:

$$T = \frac{2}{3(N_{\text{atoms}} - 1)} \times \sum_{i=1}^{N_{\text{atoms}}} \frac{m_i v_i^2}{2} \qquad (18)$$

Figure 3: Results on the Lennard-Jones system with 800 atoms and $N = 4800$. Forces on 50 atoms are used to train $\mathcal{L}_{\mathbf{x},p}$ for all the latent model structures. Each model is trained with ten repeats.

here $v_i$ is the velocity of the $i$-th atom and $m_i = 1$. We find another 31 closure variables using the autoencoder, then the latent dimension is 32. In our experiment, we also adopt the trajectory distribution of microscopic state $\mathbf{x}$ for $\mathcal{D}$. For data generation with partial forces, we choose $p = 1/16$. For each $\mathbf{x}$ we randomly choose 50 atoms for force computations.

**Results** All the models are trained with the same size of data. Fig. 3 shows the test error on 10 test trajectories. The test errors of using $\mathcal{L}_{\mathbf{x},p}(p = 1/16)$ are relatively small ($\sim 10^{-3}$), which validates the accuracy of our model on the Lennard-Jones system. It is easy to observe from Fig. 3 that for all the latent model structures, models trained with $\mathcal{L}_{\mathbf{x},p}$ can always outperform those trained with $\mathcal{L}_{\mathbf{z}}$. This validates $\mathcal{L}_{\mathbf{x},p}$ is *robust over different latent model structures*.

### 5.3 Robustness to Different Microscopic dynamics

We have already validated the accuracy and force computation efficiency of $\mathcal{L}_{\mathbf{x},p}$ on the Predatory-Prey system and the Lennard-Jones system, but their microscopic dimension is still not big enough. In this subsection, we focus on the robustness of our method to different systems including those with much larger microscopic dimension. We will consider two large systems: the Allen-Cahn system and a larger Lennard-Jones system with 51200 atoms.

**Allen-Cahn System** The Allen-Cahn equation is widely used to model the phase transition process in binary mixtures (Allen & Cahn, 1979; Del Pino et al., 2008; Shen & Yang, 2010; Kim et al., 2021; Yang et al., 2023). We consider the 2-dimensional Allen-Cahn equation with zero Neumann boundary condition on a bounded domain:

$$\partial_t v = \nabla^2 v - \frac{1}{\epsilon^2} F'(v) \text{ on } \Omega = [0,1] \times [0,1]$$
$$\partial_{\mathbf{n}} v = 0 \text{ on } \partial\Omega, \qquad (19)$$

where $v(-1 \leqslant v \leqslant 1)$ denotes the difference of the concentration of the two phases. $F(v)$ is usually chosen to be the double potential taking the form of $F(v) = \frac{1}{4}(v^2 - 1)^2$. The Allen-Cahn equation is the $L^2$ gradient flow of the free energy functional $\mathcal{E}(v) \in \mathbb{R}$ in Eq. (20) (Bartels, 2015).

$$\mathcal{E}(v) = \int_\mu \left( \frac{1}{\epsilon^2} F(v) + \frac{1}{2} \|\nabla v\|_2^2 \right) \, \mathrm{d}x \, \mathrm{d}y \qquad (20)$$

The free energy functional $\mathbf{z}^* = \mathcal{E}(v)$ is a macroscopic observable of wide interest, hence we choose $\mathcal{E}(v)$ as our target macroscopic observable.

The spatial domain is discretized into $200 \times 200$ grids, then the dimension of the microscopic system is $N = 40000$. We find another 31 closure variables using the autoencoder hence the total dimension of the macroscopic system is 32, which is much smaller compared to the dimension of the microscopic system. We consider $\mathcal{D}$ to be the trajectory distribution of $\mathbf{x}$. We choose $p = 1/25$, each time the forces on 1600 grids are calculated for training $\mathcal{L}_{\mathbf{x},p}$.

**Lennard-Jones System (large)**  To further demonstrate the capacity of our method, we scale up the Lennard-Jones system in Section 5.2 to encompass 51200 atoms, then $N = 307200$. The size of the simulation box is increased from $10 \times 10 \times 10$ to $40 \times 40 \times 40$ to keep the density unchanged.

**Results**  For a summary of the experiments and the results, we refer the reader to Table 1. Note that for the Lennard-Jones system (large), we still use the forces on 50 atoms for training, thus $p = 1/1024$. For the training of $\mathcal{L}_{\mathbf{x},p}(p = 1/1024)$, only 5000 configurations with partial forces are used due to memory limit, which is equivalent to $5000/1024 \approx 5$ training data with forces on all the atoms. Obviously, training data with size 5 is way too small, hence the results of $\mathcal{L}_{\mathbf{z}}$ are not reported for this system.

From the results shown in Table 1, one can observe that for a variety of problems, including those modeled by partial differential equations or molecular dynamics simulations, $\mathcal{L}_{\mathbf{x},p}$ can always outperform $\mathcal{L}_{\mathbf{z}}$. This shows the *robustness of our method to a variety of systems*. Moreover, the success of our method on the Lennard-Jones system (large) demonstrates the ability and efficiency of our method when scaled to very large systems.

We also compare the number of force computations that are required for models trained with $\mathcal{L}_{\mathbf{z}}$ and $\mathcal{L}_{\mathbf{x},p}$ to reach test error $e_{\text{tol}} = 3 \times 10^{-3}$. Fig. 4 shows the results on the Lennard-Jones system with different sizes. Lennard-Jones system with $800, 2700, 6400, 21600$ atoms are considered and the density is 0.8 for all the systems. The number of force computations here refers to the total number of forces on atoms that are used. For example, if $\mathcal{L}_{\mathbf{x},p}$ uses 100 configurations to train, and for each configuration, forces on 50 atoms are calculated, then the number of force computations would be $100 \times 50 = 5000$. From Fig. 4 we can observe that as the system size increases, the number of force computations required by $\mathcal{L}_{\mathbf{z}}$ continues increasing. In the experiment, we find that the number of training data does not change a lot. The increase in the number of force computations is mainly due to the system size increases,

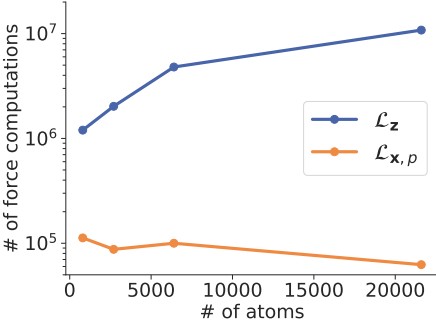

Figure 4: Number of force computations required to achieve $e_{\text{tol}} = 3 \times 10^{-3}$ on Lennard-Jones system with different sizes. Forces on 50 atoms are used to train $\mathcal{L}_{\mathbf{x},p}$ for systems of different sizes.

then for each configuration, more force computations are required. However, the number of force computations even decreases a bit. One possible explanation is that, as the system size increases, the dynamics become less fluctuating and are easy to learn.

## 6  Conclusion

We present a framework for modeling the dynamics of macroscopic observables from partial computation of microscopic forces. We theoretically and experimentally demonstrate the accuracy, force computation efficiency, and robustness of our method through different problems. Finally, we apply our method to a very large Lennard-Jones system which contains 51200 atoms.

While our method can learn the macroscopic dynamics from partial computation of microscopic forces, it relies on the sparsity assumption. For systems that do not satisfy the sparsity assumption, the calculation of partial computation of microscopic forces is not efficient, thus it is not beneficial to learn from partial forces computation. For example, in the McKean-Vlasov system, the force on

each microscopic coordinate depends on the collective behavior of all the other coordinates (Méléard, 1996).

Another limitation is the structure of the autoencoder. For particle systems such as the Lennard-Jones system, an ideal encoder should be permutation-invariant. Currently, we use MLP for the encoder, which can be improved. Additionally, our method assumes the microscopic state to be sampled from a distribution $\mathcal{D}$. We choose $\mathcal{D}$ to be trajectory distribution in the experiments, but in reality trajectory distribution of large systems may be impossible to obtain. Active learning is commonly applied to efficiently select microscopic configurations for training (Ang et al., 2021; Zhang et al., 2019; Farache et al., 2022; Kulichenko et al., 2023; Duschatko et al., 2024). It is of interest to combine active learning and our proposed method to overcome the difficulty of the choice of $\mathcal{D}$. Furthermore, our method assumes the microscopic dynamics to be deterministic, another future direction could be generalizing our method to stochastic systems.

### Acknowledgments

This research is supported by the National Research Foundation, Singapore under its AI Singapore Programme (AISG Award No: AISG3-RP-2022-028).

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

## Appendix

Let $\mathbf{x} = (\mathbf{x}_1, \cdots, \mathbf{x}_n) \in \mathcal{X} \subset \mathbb{R}^N$, $\mathbf{y} = \mathbf{f}(\mathbf{x}) \in \mathcal{Y} \subset \mathbb{R}^N$, $\boldsymbol{\theta} \in \Theta$, here $\Theta$ is the set of our model parameters. Write $\mathcal{D}^K = \{\mathbf{x}^i, \mathbf{y}^i\}_{i=1,\cdots,K}$, then $\mathcal{L}_{\mathbf{z}}(\boldsymbol{\theta}), \mathcal{L}_{\mathbf{x}}(\boldsymbol{\theta})$ can be written as:

$$
\begin{aligned}
\mathcal{L}_{\mathbf{z}}(\boldsymbol{\theta}) &= \frac{1}{K} \sum_{(\mathbf{x},\mathbf{y}) \in \mathcal{D}^K} \|\varphi'(\mathbf{x})\mathbf{y} - \mathbf{g}_{\boldsymbol{\theta}}(\mathbf{z})\|_2^2 \\
\mathcal{L}_{\mathbf{x}}(\boldsymbol{\theta}) &= \frac{1}{K} \sum_{(\mathbf{x},\mathbf{y}) \in \mathcal{D}^K} \|\mathbf{y} - (\varphi'(\mathbf{x}))^{\dagger}\mathbf{g}_{\boldsymbol{\theta}}(\mathbf{z})\|_2^2
\end{aligned}
\tag{21}
$$

In our method, we map the training procedure from the macroscopic coordinates to microscopic coordinates and use partial computation of the microscopic forces to train. We treat $\mathcal{L}_{\mathbf{z}}(\boldsymbol{\theta})$ as baseline and give detailed theoretical analysis of the possible error introduced by using $\mathcal{L}_{\mathbf{x},p}(\boldsymbol{\theta})$. The error can be controlled by two parts: (i) Convert loss from $\mathbf{z}$ space to $\mathbf{x}$ space, *i.e.*, the error introduced by using $\mathcal{L}_{\mathbf{x}}(\boldsymbol{\theta})$ (ii) Use partial labels, *i.e.*, the error between $\mathcal{L}_{\mathbf{x},p}(\boldsymbol{\theta})$ and $\mathcal{L}_{\mathbf{x}}(\boldsymbol{\theta})$.

We will analyze the first part and the second part of the error accordingly (Appendix A). More experimental details are provided in Appendix B.

## A Theoretical Analysis

### A.1 Proof of Theorem 1

**Theorem.** *Assume for any $\mathbf{x} \sim \mathcal{D}$, the eigenvalues of $\varphi'(\mathbf{x})\varphi'(\mathbf{x})^T$ are lower bounded by $b_1$ and upper bounded by $b_2$, $0 < b_1 \leqslant b_2$. Then:*

$$
b_1(\mathcal{L}_{\mathbf{x}}(\boldsymbol{\theta}) + C) \leqslant \mathcal{L}_{\mathbf{z}}(\boldsymbol{\theta}) \leqslant b_2(\mathcal{L}_{\mathbf{x}}(\boldsymbol{\theta}) + C)
\tag{22}
$$

*here $C$ does not depend on $\boldsymbol{\theta}$ hence does not affect the optimization.*

*Proof.* We can write $\varphi'(\mathbf{x})$ in the following form by leveraging singular value decomposition:

$$
\varphi'(\mathbf{x}) = \mathbf{U}\boldsymbol{\Sigma}\mathbf{V}^T,
\tag{23}
$$

where $\varphi'(\mathbf{x}) \in \mathbb{R}^{d \times N}$, $\boldsymbol{\Sigma} \in \mathbb{R}^{d \times N}$ is a rectangular diagonal matrix, $\mathbf{U} \in \mathbb{R}^{d \times d}$ and $\mathbf{V} \in \mathbb{R}^{N \times N}$ are two orthogonal matrices, $d \ll N$. $\mathbf{U}, \boldsymbol{\Sigma}, \mathbf{V}$ actually depends on $\mathbf{x}$ but for simplicity we omit the dependence in notation. During the training of the autoencoder, we enforce $\varphi'(\mathbf{x})$ to have full row rank, then the diagonal items are all nonzero, *i.e.*, $\boldsymbol{\Sigma}_{i,i} = \lambda_i(\mathbf{x}) \neq 0, i = 1, \cdots, d$. We define $\boldsymbol{\Sigma}^{\dagger} \in \mathbb{R}^{N \times d}$ to be the rectangular diagonal matrix such that the diagonal items are $(\boldsymbol{\Sigma}^{\dagger})_{i,i} = \lambda_i^{-1}(\mathbf{x}) \neq 0, i = 1, \cdots, d$ and all the remaining items are zero. Actually $\boldsymbol{\Sigma}^{\dagger}$ is the Moore-Penrose inverse of $\boldsymbol{\Sigma}$.

Then $(\varphi'(\mathbf{x}))^{\dagger}$ can be calculated by:

$$
(\varphi'(\mathbf{x}))^{\dagger} = \mathbf{V}\boldsymbol{\Sigma}^{\dagger}\mathbf{U}^T
\tag{24}
$$

If we denote the $i$-th column of $\mathbf{V}$ by $\mathbf{v}_i$, the i-th column of $\mathbf{U}$ by $\mathbf{u}_i$, then $\mathbf{V} = (\mathbf{v}_1, \cdots, \mathbf{v}_N)$, $\mathbf{U} = (\mathbf{u}_1, \cdots, \mathbf{u}_d)$, and we can rewrite $\mathcal{L}_{\mathbf{z}}(\boldsymbol{\theta})$ and $\mathcal{L}_{\mathbf{x}}(\boldsymbol{\theta})$:

$$
\begin{aligned}
\mathcal{L}_{\mathbf{z}}(\boldsymbol{\theta}) &= \frac{1}{K} \sum_{(\mathbf{x},\mathbf{y}) \in \mathcal{D}^K} \|\varphi'(\mathbf{x})\mathbf{y} - \mathbf{g}_{\boldsymbol{\theta}}(\mathbf{z})\|_2^2 \\
&= \frac{1}{K} \sum_{(\mathbf{x},\mathbf{y}) \in \mathcal{D}^K} \|\mathbf{U}\boldsymbol{\Sigma}\mathbf{V}^T\mathbf{y} - \mathbf{g}_{\boldsymbol{\theta}}(\mathbf{z})\|_2^2 \\
&= \frac{1}{K} \sum_{(\mathbf{x},\mathbf{y}) \in \mathcal{D}^K} \|\mathbf{U}\boldsymbol{\Sigma}\mathbf{V}^T\mathbf{y} - \mathbf{U}\mathbf{U}^T\mathbf{g}_{\boldsymbol{\theta}}(\mathbf{z})\|_2^2 \\
&= \frac{1}{K} \sum_{(\mathbf{x},\mathbf{y}) \in \mathcal{D}^K} (\boldsymbol{\Sigma}\mathbf{V}^T\mathbf{y} - \mathbf{U}^T\mathbf{g}_{\boldsymbol{\theta}}(\mathbf{z}))^T\mathbf{U}^T\mathbf{U}(\boldsymbol{\Sigma}\mathbf{V}^T\mathbf{y} - \mathbf{U}^T\mathbf{g}_{\boldsymbol{\theta}}(\mathbf{z})) \\
&= \frac{1}{K} \sum_{(\mathbf{x},\mathbf{y}) \in \mathcal{D}^K} \|\boldsymbol{\Sigma}\mathbf{V}^T\mathbf{y} - \mathbf{U}^T\mathbf{g}_{\boldsymbol{\theta}}(\mathbf{z})\|_2^2 \\
&= \frac{1}{K} \sum_{i=1}^d \sum_{(\mathbf{x},\mathbf{y}) \in \mathcal{D}^K} \|\lambda_i(\mathbf{x})\mathbf{v}_i^T\mathbf{y} - \mathbf{u}_i^T\mathbf{g}_{\boldsymbol{\theta}}(\mathbf{z})\|_2^2
\end{aligned}
\tag{25}
$$

$$\mathcal{L}_{\mathbf{x}}(\boldsymbol{\theta}) = \frac{1}{K}\sum_{(\mathbf{x},\mathbf{y})\in\mathcal{D}^K}\|\mathbf{y} - (\boldsymbol{\varphi}'(\mathbf{x}))^{\dagger}\mathbf{g}_{\boldsymbol{\theta}}(\mathbf{z})\|_2^2$$

$$= \frac{1}{K}\sum_{(\mathbf{x},\mathbf{y})\in\mathcal{D}^K}\|\mathbf{y} - \mathbf{V}\boldsymbol{\Sigma}^{\dagger}\mathbf{U}^T\mathbf{g}_{\boldsymbol{\theta}}(\mathbf{z})\|_2^2$$

$$= \frac{1}{K}\sum_{(\mathbf{x},\mathbf{y})\in\mathcal{D}^K}\|\mathbf{V}\mathbf{V}^T\mathbf{y} - \mathbf{V}\boldsymbol{\Sigma}^{\dagger}\mathbf{U}^T\mathbf{g}_{\boldsymbol{\theta}}(\mathbf{z})\|_2^2)$$

$$= \frac{1}{K}\sum_{(\mathbf{x},\mathbf{y})\in\mathcal{D}^K}(\mathbf{V}^T\mathbf{y} - \boldsymbol{\Sigma}^{\dagger}\mathbf{U}^T\mathbf{g}_{\boldsymbol{\theta}}(\mathbf{z}))^T\mathbf{V}^T\mathbf{V}(\mathbf{V}^T\mathbf{y} - \boldsymbol{\Sigma}^{\dagger}\mathbf{U}^T\mathbf{g}_{\boldsymbol{\theta}}(\mathbf{z})$$

$$= \frac{1}{K}\sum_{(\mathbf{x},\mathbf{y})\in\mathcal{D}^K}\|\mathbf{V}^T\mathbf{y} - \boldsymbol{\Sigma}^{\dagger}\mathbf{U}^T\mathbf{g}_{\boldsymbol{\theta}}(\mathbf{z})\|_2^2$$

$$= \frac{1}{K}\sum_{i=1}^{d}\sum_{(\mathbf{x},\mathbf{y})\in\mathcal{D}^K}\|\mathbf{v}_i^T\mathbf{y} - \lambda_i^{-1}(\mathbf{x})\mathbf{u}_i^T\mathbf{g}_{\boldsymbol{\theta}}(\mathbf{z})\|_2^2 + \frac{1}{K}\sum_{i=d+1}^{N}\sum_{(\mathbf{x},\mathbf{y})\in\mathcal{D}^K}\|\mathbf{v}_i^T\mathbf{y}\|_2^2$$

$$= \frac{1}{K}\sum_{i=1}^{d}\sum_{(\mathbf{x},\mathbf{y})\in\mathcal{D}^K}\lambda_i^{-2}(\mathbf{x})\|\lambda_i(\mathbf{x})\mathbf{v}_i^T\mathbf{y} - \mathbf{u}_i^T\mathbf{g}_{\boldsymbol{\theta}}(\mathbf{z})\|_2^2 + \frac{1}{K}\sum_{i=d+1}^{N}\sum_{(\mathbf{x},\mathbf{y})\in\mathcal{D}^K}\|\mathbf{v}_i^T\mathbf{y}\|_2^2 \tag{26}$$

We define:

$$\hat{\mathcal{L}}_{\mathbf{x}}(\boldsymbol{\theta}) = \frac{1}{K}\sum_{i=1}^{d}\sum_{(\mathbf{x},\mathbf{y})\in\mathcal{D}^K}\lambda_i^{-2}(\mathbf{x})\|\lambda_i(\mathbf{x})\mathbf{v}_i^T\mathbf{y} - \mathbf{u}_i^T\mathbf{g}_{\boldsymbol{\theta}}(\mathbf{z})\|_2^2$$

$$C = -\frac{1}{K}\sum_{i=d+1}^{N}\sum_{(\mathbf{x},\mathbf{y})\in\mathcal{D}^K}\|\mathbf{v}_i^T\mathbf{y}\|_2^2 \tag{27}$$

then $\mathcal{L}_{\mathbf{x}}(\boldsymbol{\theta}) = \hat{\mathcal{L}}_{\mathbf{x}}(\boldsymbol{\theta}) - C$, $C$ does not depend on $\boldsymbol{\theta}$ and:

$$\min_{\boldsymbol{\theta}}\mathcal{L}_{\mathbf{x}}(\boldsymbol{\theta}) \iff \min_{\boldsymbol{\theta}}\hat{\mathcal{L}}_{\mathbf{x}}(\boldsymbol{\theta}). \tag{28}$$

Comparing $\mathcal{L}_{\mathbf{z}}(\boldsymbol{\theta})$ and $\hat{\mathcal{L}}_{\mathbf{x}}(\boldsymbol{\theta})$, we observe that the only difference between $\mathcal{L}_{\mathbf{z}}(\boldsymbol{\theta})$ and $\hat{\mathcal{L}}_{\mathbf{x}}(\boldsymbol{\theta})$ is that for every term $\|\lambda_i(\mathbf{x})\mathbf{v}_i^T\mathbf{f}(\mathbf{x}) - \mathbf{u}_i^T\mathbf{g}_{\boldsymbol{\theta}}(\mathbf{z})\|_2^2$, there is a constant $\lambda_i^{-2}(\mathbf{x})$ multiplied to it. Hence $\hat{\mathcal{L}}_{\mathbf{x}}(\boldsymbol{\theta})$ is a weighted version of $\mathcal{L}_{\mathbf{z}}(\boldsymbol{\theta})$. Note that if the eigenvalues of $\boldsymbol{\varphi}'(\mathbf{x})$ are $\lambda_i(\mathbf{x}), i = 1, \cdots, d$, then the eigenvalues of $\boldsymbol{\varphi}'(\mathbf{x})\boldsymbol{\varphi}'(\mathbf{x})^T$ are $\lambda_i^2(\mathbf{x}), i = 1, \cdots, d$.

Since the eigenvalues of $\boldsymbol{\varphi}'(\mathbf{x})\boldsymbol{\varphi}'(\mathbf{x})^T$ are lower bounded by $b_1 > 0$ and upper bounded by $b_2$, $i.e.$, $\forall \mathbf{x} \in \mathcal{X}, 0 < b_1 \leqslant \lambda_i^2(\mathbf{x}) \leqslant b_2$, then:

$$b_2^{-1}\mathcal{L}_{\mathbf{z}}(\boldsymbol{\theta}) \leqslant \mathcal{L}_{\mathbf{x}}(\boldsymbol{\theta}) + C \leqslant b_1^{-1}\mathcal{L}_{\mathbf{z}}(\boldsymbol{\theta}) \tag{29}$$

or equivalently,

$$b_1(\mathcal{L}_{\mathbf{x}}(\boldsymbol{\theta}) + C) \leqslant \mathcal{L}_{\mathbf{z}}(\boldsymbol{\theta}) \leqslant b_2(\mathcal{L}_{\mathbf{x}}(\boldsymbol{\theta}) + C) \tag{30}$$

$\square$

By minimizing $\hat{\mathcal{L}}_{\mathbf{x}}(\boldsymbol{\theta})$, we are actually narrowing the region of $\mathcal{L}_{\mathbf{z}}(\boldsymbol{\theta})$. Hence we want $b_1$ and $b_2$ to be as close as possible. In the extreme case where $b_1 = b_2$, minimizing $\hat{\mathcal{L}}_{\mathbf{x}}(\boldsymbol{\theta})$ is just equivalent to minimizing $\mathcal{L}_{\mathbf{z}}(\boldsymbol{\theta})$. Another observation is that $\hat{\mathcal{L}}_{\mathbf{x}}(\boldsymbol{\theta})$ is a weighted version of $\mathcal{L}_{\mathbf{x}}(\boldsymbol{\theta})$, and if there exists $i$ such that $\lambda_i(\mathbf{x})^2$ is too small compared to the others, the weighted sum will be dominated by it in Eq. (27).

The above insights guide us to constrain the condition number of $\boldsymbol{\varphi}'(\mathbf{x})\boldsymbol{\varphi}'(\mathbf{x})^T$ during the training of autoencoder, $i.e.$, we require $\boldsymbol{\varphi}'(\mathbf{x})\boldsymbol{\varphi}'(\mathbf{x})^T$ to be well-conditioned through Eq. (6).

### A.2 Proof of Eq. (13)

$$\mathbb{E}_{\mathbf{x}^1,\cdots,\mathbf{x}^K}\mathbb{E}_{\mathbf{I}(\mathbf{x}^1),\cdots,\mathbf{I}(\mathbf{x}^K)}\mathcal{L}_{\mathbf{x},p}(\boldsymbol{\theta}) = \mathbb{E}\big[\frac{1}{pK}\sum_{i=1}^{K}\|\mathbf{f}_{\mathbf{I}(\mathbf{x}^i)}(\mathbf{x}^i) - (\boldsymbol{\varphi}'(\mathbf{x}^i))^{\dagger}_{\mathbf{I}(\mathbf{x}^i)}\mathbf{g}_{\boldsymbol{\theta}}(\mathbf{z}^i)\|_2^2\big]$$

$$= \frac{1}{p}\mathbb{E}_{\mathbf{x}}\mathbb{E}_{\mathbf{I}(\mathbf{x})}\big[\|\mathbf{f}_{\mathbf{I}(\mathbf{x})}(\mathbf{x}) - (\boldsymbol{\varphi}'(\mathbf{x}))^{\dagger}_{\mathbf{I}(\mathbf{x})}\mathbf{g}_{\boldsymbol{\theta}}(\mathbf{z})\|_2^2\big]$$

$$= \frac{1}{p}\mathbb{E}_{\mathbf{x}}\mathbb{E}_{\mathbf{I}(\mathbf{x})}\big[\sum_{i=1}^{n}\mathbf{I}_i(\mathbf{x})\cdot\|\mathbf{f}_i(\mathbf{x}) - (\boldsymbol{\varphi}'(\mathbf{x}))^{\dagger}_i\mathbf{g}_{\boldsymbol{\theta}}(\mathbf{z})\|_2^2\big]$$

$$= \frac{1}{p}\mathbb{E}_{\mathbf{x}}\big[\sum_{i=1}^{n}\mathbb{E}_{\mathbf{I}(\mathbf{x})}\mathbf{I}_i(\mathbf{x})\cdot\|\mathbf{f}_i(\mathbf{x}) - (\boldsymbol{\varphi}'(\mathbf{x}))^{\dagger}_i\mathbf{g}_{\boldsymbol{\theta}}(\mathbf{z})\|_2^2\big] \tag{31}$$

$$= \mathbb{E}_{\mathbf{x}}\big[\sum_{i=1}^{n}\|\mathbf{f}_i(\mathbf{x}) - (\boldsymbol{\varphi}'(\mathbf{x}))^{\dagger}_i\mathbf{g}_{\boldsymbol{\theta}}(\mathbf{z})\|_2^2\big]$$

$$= \mathbb{E}_{\mathbf{x}}\big[\|\mathbf{f}(\mathbf{x}) - (\boldsymbol{\varphi}'(\mathbf{x}))^{\dagger}\mathbf{g}_{\boldsymbol{\theta}}(\mathbf{z})\|_2^2\big]$$

$$= \mathbb{E}_{\mathbf{x}^1,\cdots,\mathbf{x}^K}\mathcal{L}_{\mathbf{x}}(\boldsymbol{\theta})$$

## A.3 Proof of Theorem 2

In this section we will prove the behavior of the minimizer found by $\mathcal{L}_{\mathbf{x},p}$ in the limit. Our proof relies on the statistical learning theory and especially Rademacher complexity. We will provide some background information first.

Let $\mathcal{H}$ be a family of real-valued functions with domain $\mathcal{W}$ and integrable w.r.t. $\mathbb{P}$, here $\mathbb{P}$ is a probability over $\mathcal{W}$. $\mathcal{W}^n = (\mathbf{w}^1, \cdots, \mathbf{w}^n)$ is a collection of i.i.d. samples from probability distribution $\mathcal{P}$ defined over $\mathcal{W}$.

We will use the tool of Rademacher complexity:

**Definition 1.** *Let $\mathcal{H}, \mathcal{W}^n, \mathbb{P}$ be defined as before. The empirical Rademacher complexity of $\mathcal{H}$ with respect to $\mathcal{W}^n$ is defined as:*

$$\mathcal{R}_{\mathcal{W}^n}(\mathcal{H}) = \mathbb{E}_{\boldsymbol{\sigma}} \left[ \sup_{h \in \mathcal{H}} \frac{1}{n} \sum_{i=1}^n \sigma_i h(\mathbf{w}^i) \right], \tag{32}$$

*$\boldsymbol{\sigma} = (\sigma_1, \cdots, \sigma_n), \{\sigma_i\}_{i=1}^n$ are independent random variables uniformly chosen from $\{-1, 1\}$, with $P(\sigma_i = 1) = P(\sigma_i = -1) = 0.5$. Taking the expectation with respect to $\mathcal{W}^n$ yields the Rademacher complexity of the functional class $\mathcal{H}$:*

$$\mathcal{R}_n(\mathcal{H}) = \mathbb{E}_{\mathcal{W}^n} \mathbb{E}_{\boldsymbol{\sigma}} \left[ \sup_{h \in \mathcal{H}} \frac{1}{n} \sum_{i=1}^n \sigma_i h(\mathbf{w}^i) \right]. \tag{33}$$

Then one can derive the generalization bound in terms of the Rademacher complexity (Wainwright, 2019):

**Theorem 3.** *Assume $\mathcal{H}$ is uniformly bounded by $b$ (i.e., $\|f\|_\infty \leqslant b$. Then for all $n \geqslant 1$ and $\delta \geqslant 0$, we have*

$$\sup_{h \in \mathcal{H}} \left| \frac{1}{n} \sum_{i=1}^n h(\mathbf{w}^i) - \mathbb{E}[h] \right| \leqslant 2\mathcal{R}_n(\mathcal{H}_{\mathbf{x}}) + \delta \tag{34}$$

*with probability at least $1 - 2\exp\left(-\frac{n\delta^2}{8b^2}\right)$. Consequently, as long as $\mathcal{R}_n(\mathcal{H}_{\mathbf{x}}) = o(1)$, we have $\frac{1}{n} \sum_{i=1}^n f(X_i) - \mathbb{E}[f] \xrightarrow{a.s.} 0, \forall f \in \mathcal{H}_{\mathbf{x}}$.*

Now in our problem, let $\mathcal{H}_{\mathbf{x},p}$ be the following one function class:

$$\mathcal{H}_{\mathbf{x},p} = \{h_{\boldsymbol{\theta},p} : \mathcal{X} \times \mathcal{Y} \times \mathcal{I} \to \mathbb{R}; h_{\boldsymbol{\theta},p}(\mathbf{x}, \mathbf{y}, \mathrm{I}(\mathbf{x})) = \tfrac{1}{p}\|\mathbf{y}_{\mathrm{I}(\mathbf{x})} - (\boldsymbol{\varphi}'(\mathbf{x}))^\dagger_{\mathrm{I}(\mathbf{x})} \mathbf{g}_{\boldsymbol{\theta}}(\mathbf{z})\|_2^2, \boldsymbol{\theta} \in \Theta\} \tag{35}$$

Then

$$\mathcal{L}_{\mathbf{x},p}(\boldsymbol{\theta}) = \frac{1}{K} \sum_{(\mathbf{x},\mathbf{y}) \in \mathcal{D}^K} h_{\boldsymbol{\theta},p}(\mathbf{x}, \mathbf{y}, \mathrm{I}(\mathbf{x})) \tag{36}$$

Now we can prove Theorem 2:

**Theorem.** *Let $\tilde{\mathcal{L}}_{\mathbf{x}}(\boldsymbol{\theta}) = \mathbb{E}\mathcal{L}_{\mathbf{x}}(\boldsymbol{\theta}), \boldsymbol{\theta}^* \in \arg\min_{\boldsymbol{\theta}} \tilde{\mathcal{L}}_{\mathbf{x}}(\boldsymbol{\theta}), \boldsymbol{\theta}_{K,p} \in \arg\min_{\boldsymbol{\theta}} \mathcal{L}_{\mathbf{x},p}(\boldsymbol{\theta})$, if $\mathcal{H}_{\mathbf{x},p}$ is uniformly bounded by $b_{\mathbf{x},p}$ and $\mathcal{R}_K(\mathcal{H}_{\mathbf{x},p}) = o(1)$, then:*

$$\tilde{\mathcal{L}}_{\mathbf{x}}(\boldsymbol{\theta}_{K,p}) - \tilde{\mathcal{L}}_{\mathbf{x}}(\boldsymbol{\theta}^*) \xrightarrow{a.s.} 0 \tag{37}$$

*Proof.* We define $\tilde{\mathcal{L}}_{\mathbf{x},p}(\boldsymbol{\theta}) = \mathbb{E}\mathcal{L}_{\mathbf{x},p}(\boldsymbol{\theta})$, then $\tilde{\mathcal{L}}_{\mathbf{x}} = \tilde{\mathcal{L}}_{\mathbf{x},p}$ by Appendix A.2. Applying Theorem 3, we get

$$\sup_{\boldsymbol{\theta} \in \Theta} \left| \mathcal{L}_{\mathbf{x},p}(\boldsymbol{\theta}) - \tilde{\mathcal{L}}_{\mathbf{x},p}(\boldsymbol{\theta}) \right| \leqslant 2\mathcal{R}_K(\mathcal{H}_{\mathbf{x},p}) + \delta \tag{38}$$

with probability at least $1 - 2\exp(-\frac{K\delta^2}{8b_{\mathbf{x},p}})$, and $\mathcal{L}_{\mathbf{x},p}(\boldsymbol{\theta}) \xrightarrow{a.s.} \tilde{\mathcal{L}}_{\mathbf{x},p}(\boldsymbol{\theta}), \forall \boldsymbol{\theta} \in \Theta$.

Note that $\boldsymbol{\theta}^* \in \arg\min_{\boldsymbol{\theta}} \tilde{\mathcal{L}}_{\mathbf{x}}(\boldsymbol{\theta}) \in \arg\min_{\boldsymbol{\theta}} \tilde{\mathcal{L}}_{\mathbf{x},p}(\boldsymbol{\theta})$, then

$$0 \leqslant \tilde{\mathcal{L}}_{\mathbf{x}}(\boldsymbol{\theta}_{K,p}) - \tilde{\mathcal{L}}_{\mathbf{x}}(\boldsymbol{\theta}^*) = \underbrace{\tilde{\mathcal{L}}_{\mathbf{x},p}(\boldsymbol{\theta}_{K,p}) - \mathcal{L}_{\mathbf{x},p}(\boldsymbol{\theta}_{K,p})}_{\xrightarrow{a.s.} 0} + \underbrace{\mathcal{L}_{\mathbf{x},p}(\boldsymbol{\theta}_{K,p}) - \mathcal{L}_{\mathbf{x},p}(\boldsymbol{\theta}^*)}_{\leqslant 0}$$
$$+ \underbrace{\mathcal{L}_{\mathbf{x},p}(\boldsymbol{\theta}^*) - \tilde{\mathcal{L}}_{\mathbf{x},p}(\boldsymbol{\theta}^*)}_{\xrightarrow{a.s.} 0} \tag{39}$$

thus $\tilde{\mathcal{L}}_{\mathbf{x},p}(\boldsymbol{\theta}_{K,p}) - \tilde{\mathcal{L}}_{\mathbf{x},p}(\boldsymbol{\theta}^*) \xrightarrow{a.s.} 0$. $\qquad\square$

# B   Experiment Details

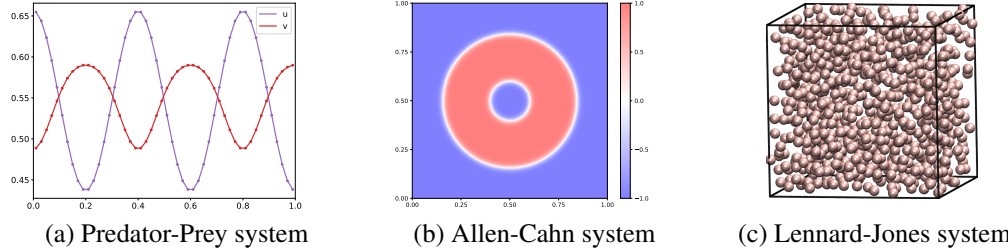

|            (a) Predator-Prey system | (b) Allen-Cahn system | (c) Lennard-Jones system |

Figure 5: Visualization of the microscopic state of each system

## B.1   Predator-Prey System

We consider the Neumann boundary condition $\partial_n u = 0, \partial_n v = 0$ on $\partial\Omega$ and the following initial conditions

$$u(x,0) = \mu + \sigma \cos(5\pi x)$$
$$v(x,0) = 1 - \mu - \sigma \cos(5\pi x), \quad (\mu,\sigma) \in [0,0.2] \times [0.4,0.6] \tag{40}$$

The Neumann boundary condition is commonly used in mathematical models of ecosystems, restricting any movement of the species out of the boundary.

We approximate the spatial derivatives in Eq. (15) with finite difference method:

$$\frac{\partial^2 v}{\partial x^2}(x_i, t) \approx \frac{v(x_{i+1}, t) - 2v(x_i, t) + v(x_{i-1}, t)}{\Delta x^2} \quad 2 \leqslant i \leqslant 49$$
$$\frac{\partial^2 v}{\partial x^2}(x_1, t) \approx \frac{v(x_2, t) - v(x_1, t)}{\Delta x^2} \tag{41}$$
$$\frac{\partial^2 v}{\partial x^2}(x_{50}, t) \approx \frac{v(x_{49}, t) - v(x_{50}, t)}{\Delta x^2}$$

Let $h_u(u,v) = u(1-u-v), h_v(u,v) = av(u-b)$, and $h_u(\mathbf{u}, \mathbf{v}), h_v(\mathbf{u}, \mathbf{v})$ denote the element-wise application of $h_u, h_v$ to each $u(x_i, t), v(x_i, t), 1 \leqslant i \leqslant 50$. Then

$$\frac{d\mathbf{u}}{dt} = h_u(\mathbf{u}, \mathbf{v})$$
$$\frac{d\mathbf{v}}{dt} = h_v(\mathbf{u}, \mathbf{v}) + \mathbf{A}\mathbf{v} \tag{42}$$

here $\mathbf{A} \in \mathbb{R}^{50 \times 50}$ is a matrix defined according to Eq. (42). Hence the predator-prey system after spatial discretization can be written in the form of Eq. (1).

In our experiment, we choose $a = 3, b = 0.4, \lambda = 0$. The training parameter set $\mathcal{T}_{\text{train}}$ of pairs $(\mu, \sigma)$ are sampled uniformly from $[0, 0.2] \times [0.4, 0.6]$. For testing, $\mathcal{T}_{\text{test}}$ is also sampled uniformly from $[0, 0.2] \times [0.4, 0.6]$, but with a different random seed from $\mathcal{T}_{\text{train}}$. The mean relative error is defined as :

$$e(\mathcal{T}_{\text{test}}) = \frac{1}{|\mathcal{T}_{\text{test}}|} \sum_{(\mu,\sigma) \in \mathcal{T}_{\text{test}}} \left( \frac{\sum_n \|\mathbf{z}^*_{\text{true}}(t_n; \mu, \sigma) - \mathbf{z}^*_{\text{pred}}(t_n; \mu, \sigma)\|_2^2}{\sum_n \|\mathbf{z}^*_{\text{true}}(t_n; \mu, \sigma)\|_2^2} \right), \tag{43}$$

here we use $\mathbf{z}(\cdot; \mu, \sigma)$ to denote the dependency of the solution on the initial condition.

Table 2: Results on the Predator-Prey system. Models are trained with different training metric $\mathcal{L}_\mathbf{x}, \mathcal{L}_{\mathbf{x},p}(p = 3/4, 1/2, 1/4, 1/5)$. Mean and standard deviation are reported over three repeats.

| # of training data | $\mathcal{L}_\mathbf{x}$ | $\mathcal{L}_{\mathbf{x},p}(p = 3/4)$ | $\mathcal{L}_{\mathbf{x},p}(p = 1/2)$ | $\mathcal{L}_{\mathbf{x},p}(p = 1/4)$ | $\mathcal{L}_{\mathbf{x},p}(p = 1/5)$ |
|---|---|---|---|---|---|
| $6.0 \times 10^2$ | $1.09_{\pm 0.57} \times 10^{-2}$ | $1.79_{\pm 1.11} \times 10^{-2}$ | $9.22_{\pm 4.46} \times 10^{-3}$ | $\mathbf{3.50}_{\pm 1.33} \times \mathbf{10^{-3}}$ | $4.00_{\pm 1.78} \times 10^{-3}$ |
| $1.5 \times 10^3$ | $4.70_{\pm 0.46} \times 10^{-3}$ | $4.56_{\pm 1.97} \times 10^{-3}$ | $3.30_{\pm 1.60} \times 10^{-3}$ | $2.19_{\pm 0.57} \times 10^{-3}$ | $\mathbf{1.92}_{\pm 0.47} \times \mathbf{10^{-3}}$ |
| $3.0 \times 10^3$ | $2.90_{\pm 1.63} \times 10^{-3}$ | $2.07_{\pm 0.41} \times 10^{-3}$ | $1.57_{\pm 0.45} \times 10^{-3}$ | $\mathbf{1.23}_{\pm 0.16} \times \mathbf{10^{-3}}$ | $1.34_{\pm 0.20} \times 10^{-3}$ |
| $6.0 \times 10^3$ | $1.24_{\pm 0.16} \times 10^{-3}$ | $1.07_{\pm 0.18} \times 10^{-3}$ | $1.46_{\pm 0.59} \times 10^{-3}$ | $\mathbf{9.08}_{\pm 0.25} \times \mathbf{10^{-4}}$ | $1.13_{\pm 0.31} \times 10^{-3}$ |
| $1.2 \times 10^4$ | $9.20_{\pm 2.30} \times 10^{-4}$ | $8.13_{\pm 0.64} \times 10^{-4}$ | $8.23_{\pm 0.13} \times 10^{-4}$ | $\mathbf{7.66}_{\pm 3.14} \times \mathbf{10^{-4}}$ | $8.36_{\pm 1.94} \times 10^{-4}$ |
| $2.4 \times 10^4$ | $6.76_{\pm 0.93} \times 10^{-4}$ | $6.85_{\pm 0.95} \times 10^{-4}$ | $6.98_{\pm 1.20} \times 10^{-4}$ | $\mathbf{5.64}_{\pm 0.18} \times \mathbf{10^{-4}}$ | $5.70_{\pm 1.00} \times 10^{-4}$ |

**Data Generation**   The microscopic equation is solved with a uniform time step $\Delta t = 0.01$ from $t = 0$ to $t = 30$ using the Euler method. We subsampled every tenth snapshot for training. During testing, the microscopic evolution equation is solved with the same $\Delta t = 0.01$ using Runge-Kutta 4-order RK4 solver. Then we encode the microscopic trajectories to obtain the ground truth latent trajectories. The predicted latent trajectories are obtained by encoding the initial microscopic state first, then solved using RK4 solver with $\Delta t = 0.1, \Delta = 0.5$ on $[0, 30]$. Fig. 6 and Fig. 7 show the true and predicted trajectories.

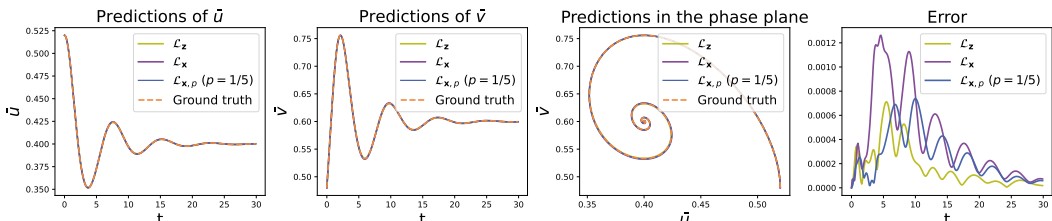

Figure 6: Latent trajectories with initial condition $\mu = 0.02, \sigma = 0.52$ and $\Delta t = 0.1$ in the Predator-Prey system.

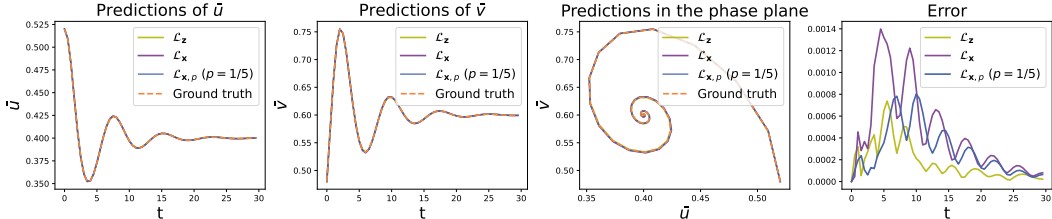

Figure 7: Latent trajectories with initial condition $\mu = 0.02, \sigma = 0.52$ and $\Delta t = 0.5$ in the Predator-Prey system.

### B.2    Allen-Cahn System

In our experiment, we consider the initial condition of a torus  (Kim et al., 2021):

$$v(x, y, 0) = -1 + \tanh\left(\frac{r_1 - d(x,y)}{\sqrt{2}\epsilon}\right) - \tanh\left(\frac{r_2 - d(x,y)}{\sqrt{2}\epsilon}\right) \tag{44}$$

here $d(x,y) = \sqrt{(x - 0.5)^2 + (y - 0.5)^2}$, $r_1 \in [0.3, 0.4]$ is the circumscribed circle radius and $r_2 \in [0.1, 0.15]$ is the inscribed circle radius. The initial condition is visualized in Fig. 5 (b).

The free energy in Eq. (20) also tends to decrease with time, following the energy dissipation law in Eq. (45). Then the minimization of the free energy drives the evolution of the system towards equilibrium.

$$\frac{\partial \mathcal{E}(v)}{\partial t} = -\int_\mu \|\partial_t v\|_2^2 \, \mathrm{d}x \, \mathrm{d}y \tag{45}$$

**Data Generation**   For both training and testing, the microscopic evolution law is solved using RK4 method with $\Delta t = 1/N = 2.5 \times 10^{-5}$ from $t = 0$ to $t = \min(t_f, 1)$. Here $t_f$ is the time when the Allen-Cahn system reaches the equilibrium. We subsample every hundredth snapshots for training. We choose $\epsilon$ in Eq. (19) and Eq. (44) to be $\frac{10 \times 200}{2\sqrt{2} \tanh^{-1}(0.9)}$ as in (Kim et al., 2021).

For testing, 50 parameter points $(r_1, r_2)$ are chosen uniformly from $[0.3, 0.4] \times [0.1, 0.15]$, and we report the mean relative error. The test parameter set $\mathcal{T}_{\text{test}}$ contains 50 parameter points $(r_1, r_2)$ are chosen uniformly from $[0.3, 0.4] \times [0.1, 0.15]$, but are sampled with a different random seed.

### B.3   Lennard-Jones System

We consider Lennard-Jones systems containing different atoms in this paper: $N_{\text{atoms}} = 800, 2700, 6400, 21600, 51200$. We use periodic boundary conditions and fix the density to be $0.8$, then the corresponding box side lengths are $10, 15, 20, 30, 40$. We simulate the Lennard-Jones system under the NVE ensemble using the LAMMPS (Thompson et al., 2022). In our experiment, $\epsilon_{ij} = \sigma_{ij} = 1, \forall i, j$, and $r_{\text{cut}} = 2.5$. The integration step is $0.001$ and each trajectory is integrated for 250 steps. We sample the initial temperature randomly from $[0.5, 1.5]$. The initial velocities are then sampled from the Maxwell–Boltzmann distribution. For each system, the initial configuration has the same atom positions and velocity direction. For testing, the initial temperatures are also randomly sampled from $[0.5, 1.5]$ but with a different random seed to the training data.

## C   Implementation Details

All the experiments are run on a single NVIDIA GeForce RTX 3090 GPU. For all the experiments, we use the multilayer perceptron (MLP) for both the encoder and the decoder. The autoencoders are trained with $\mathcal{L}_{\text{AE}}$ in Eq. (7). The condition number is the maximal eigenvalue $\lambda_{\max}$ divided by the minimal eigenvalue $\lambda_{\min}$ of $\varphi'(\mathbf{x})\varphi'(\mathbf{x})^T$:

$$\kappa\left(\varphi'(\mathbf{x})\varphi'(\mathbf{x})^T\right) = \frac{|\lambda_{\max}(\varphi'(\mathbf{x})\varphi'(\mathbf{x})^T)|}{|\lambda_{\min}(\varphi'(\mathbf{x})\varphi'(\mathbf{x})^T)|} \geqslant 1 \tag{46}$$

Since $\varphi'(\mathbf{x}) \in \mathbb{R}^{d \times N}, \varphi'(\mathbf{x})\varphi'(\mathbf{x})^T \in \mathbb{R}^{d \times d}$, and $d$ is small, the condition number $\kappa(\varphi'(\mathbf{x})\varphi'(\mathbf{x})^T)$ can be calculated efficiently. In our experiments, we calculate $\lambda_{\max}$ and $\lambda_{\min}$ with `torch.linalg.svd`. To better compare $\mathcal{L}_{\mathbf{x}}$ and $\mathcal{L}_{\mathbf{x},p}$, once finish the training of the autoencoders, we freeze them and use the encoder for macroscopic dynamics identification. For the macroscopic dynamics identification, MLP and GFINNs are used for the latent model in Section 5.2. For the rest experiments, we adopt the structure of OnsagerNet to enhance the stability for latent dynamics prediction for $\mathbf{g}_\theta$ (Yu et al., 2021).

## D   Additional Experiments

### D.1   Loss Curve of $\mathcal{L}_{\mathbf{z}}, \mathcal{L}_{\mathbf{x}}, \mathcal{L}_{\mathbf{x},p}$

To give the readers a better idea of the behaviors of the loss $\mathcal{L}_{\mathbf{z}}, \mathcal{L}_{\mathbf{x}}, \mathcal{L}_{\mathbf{x},p}(p = 1/4)$ trained on the same number of training data. Fig. 8 shows the training and test loss curve of different training metrics. Note that the training metrics of these models are different, but they are tested with the same metric Eq. (43).

### D.2   Ablation Analysis of $\lambda_{\text{cond}}$

To evaluate the influence of the hyperparameter $\lambda_{\text{cond}}$ on the performance of the loss $\mathcal{L}_{\mathbf{x},p}$, we conduct experiments with different values of $\lambda_{\text{cond}}$ and show the test error in Table 3.

From Table 3 we can observe when $\lambda_{\text{cond}}$ increases from 0 to $10^{-6}$, the test error gradually decrease. When $\lambda_{\text{cond}}$ further increases from $10^{-6}$ to $10^{-2}$, the test error gradually decrease. Among all the $\lambda_{\text{cond}}$ that we tried, the test error has the minimal value when $\lambda_{\text{cond}} = 10^{-6}$.

Theoretically, if $\lambda_{\text{cond}}$ is too low, since $\mathcal{L}_{\text{AE}} = \mathcal{L}_{\text{rec}} + \lambda_{\text{cond}}\mathcal{L}_{\text{cond}}$, there may not have enough constraint on $\mathcal{L}_{\text{cond}}$ and the condition number of $\varphi'(\mathbf{x})\varphi'(\mathbf{x})^T$ may be very large. By Theorem 1,

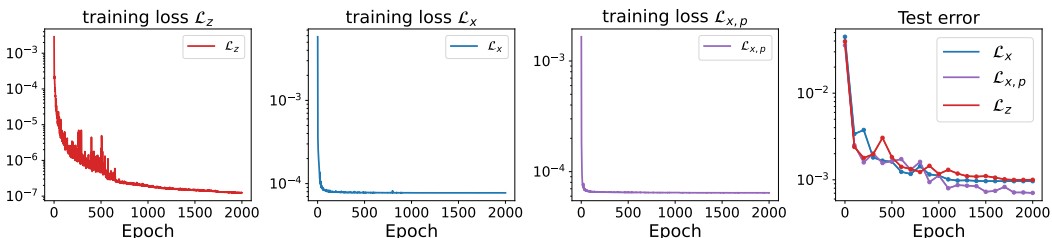

Figure 8: Loss curve of the $\mathcal{L}_\mathbf{z}, \mathcal{L}_\mathbf{x}, \mathcal{L}_{\mathbf{x},p}(p=1/4)$ on the Predator-Prey system. Models are trained with different loss functions $\mathcal{L}_\mathbf{z}, \mathcal{L}_\mathbf{x}, \mathcal{L}_{\mathbf{x},p}(p=1/4)$ on the same number of training data.

Table 3: Results on the Predator-Prey system. We train the autoencoder with different $\lambda_{\text{cond}}$ and then train the macroscopic dynamics model with loss $\mathcal{L}_{\mathbf{x},p}(p=1/5)$. The mean relative error of the macroscopic dynamics model is reported over three repeats.

| $\lambda_{\text{cond}}$ | 0 | $10^{-8}$ | $10^{-7}$ | $10^{-6}$ | $10^{-5}$ | $10^{-4}$ | $10^{-2}$ |
|---|---|---|---|---|---|---|---|
| Test error of $\mathcal{L}_{\mathbf{x},p}(p=1/5)$ | $6.42 \times 10^{-2}$ | $2.62 \times 10^{-3}$ | $2.84 \times 10^{-3}$ | $\mathbf{8.36 \times 10^{-4}}$ | $3.66 \times 10^{-3}$ | $2.60 \times 10^{-3}$ | $4.24 \times 10^{-3}$ |

a small condition number of $\varphi'(\mathbf{x})\varphi'(\mathbf{x})^T$ can guarantee the effectiveness of $\mathcal{L}_\mathbf{x}$. But when the condition number is large, there is no guarantee that $\mathcal{L}_\mathbf{x}$ thus $\mathcal{L}_{\mathbf{x},p}$ can perform well. If instead $\lambda_{\text{cond}}$ is too large, $\mathcal{L}_{\text{AE}}$ will be dominated by $\mathcal{L}_{\text{cond}}$. Then the autoencoder may not reconstruct the microscopic dynamics well and, hence may not capture the closure terms well. If the latent space is not closed enough, we can not learn the macroscopic dynamics well.

