# OpenReview forum: "Learning Macroscopic Dynamics from Partial Microscopic Observations"
_NeurIPS.cc/2024/Conference — NeurIPS 2024 poster_

### Official Review · Reviewer_RRQb · 2024-07-09

**Soundness:** 3
**Presentation:** 3
**Contribution:** 3
**Rating:** 7
**Confidence:** 3

**Summary:**

This paper introduces an efficient framework for learning macroscopic dynamics of complex systems. The training data consists of microscopic configurations with only partial dynamics, i.e., time derivatives of only a small subset of the microscopic variables. The paper shows how this information nevertheless allows unbiased estimation of dynamics at the level of low-dimensional macroscopic dynamics. Experiments show the data-efficiency of the scheme compared to learning on full microscopic dynamics.

**Strengths:**

* The paper’s approach is novel, addresses an important bottleneck in the field, and has the potential to become an important contribution in the area of ML for dynamical systems
* The paper is generally well written and the exposition is easy to follow.
* The experiments cover a diversity of dynamical systems, including MD and PDE systems.

**Weaknesses:**

* Although the theoretical results are nice and the experimental results empirically support them, I am still skeptical of the efficacy of $L_x$ as a proxy loss for $L_z$. To bolster the claims, I strongly recommend that authors present some empirical analysis on the spectrum of the Jacobian $\psi’$, or even better, plot loss curves over the course of a training run showing the behavior of $L_z$ versus $L_x$. This will give the reader a much better intuition for how these different loss functions behave.
* Without an understanding of how significant the final test errors are, it is not particularly insightful to merely show test errors after an arbitrarily selected amount of training data. Instead it would be nice to see scaling plots like in Figure 2 for the other systems in the analysis.
* Table 1 and Figure 2 are redundant and I would recommend eliminating the former.
* Theorem 2 could be further clarified. What is $\arg \min L_{x,p}$ since $L_{x,p}$ is also a random variable?
* Is it difficult or tricky to tune the weight of the conditioning number loss term? What happens if the weight on this term is not high enough? What if it is too high? Ablation analyses of these factors would improve the paper.

**Questions:**

See above

**Limitations:**

Yes

---

> ### Author Rebuttal · Authors · 2024-08-07
>
> Thank you for your detailed review and thoughtful feedback. Below are our responses:
>
> **Q: Plot loss curves over the course of a training run showing the behavior of $L_x$ versus $L_z$. This will give the reader a much better intuition for how these different loss functions behave.**
>
> A: Thanks for this suggestion. We have attached a pdf file in the top-level Author Rebuttal. In figure 1 of the pdf file, we plotted the training loss of $L_z$ and $L_x$, and their test metric. We observe that during the training of the model using $L_x$, the loss $L_x$ exhibits a sharp decline in the first several epochs. Then the loss $L_x$ exhibits a very slight decline in the following epochs, while the test error continues decreasing.
>
>
> **Q:Without an understanding of how significant the final test errors are, it is not particularly insightful to merely show test errors after an arbitrarily selected amount of training data. Instead it would be nice to see scaling plots like in Figure 2 for the other systems in the analysis.**
>
> A:  In Figure 3, each model is trained with the same number of force computations. For each latent structure, we compare the relative performance of $L_{x, p}$ versus $L_{z}$, and $L_{x, p}$ always outperforms $L_{z}$. Thus we show our model is agnostic to different latent structures. Due to the time constraint, we find it difficult to finish the scaling plots. We will continue to refine our paper based on your valuable suggestion.
>
> **Q: Table 1 and Figure 2 are redundant**
>
> A: Thanks for pointing this out. We have revised the paper to remove Table 1 from the main content.
>
> **Q: Theorem 2 could be further clarified. What is $\arg \min L_{x,p}$ since $L_{x,p}$ is also a random variable?**
>
> A: In statistical learning theory, the empirical risk is considered a random variable. Denote the data by $X$ and assume $X \sim \mathcal{D}$. Let $X^n=$ {$X_1 , \cdots, X_n$} be a collection of n i.i.d. samples from $\mathcal{D}$. Then the empirical risk is :
> $$
> L_n(\theta) = \frac{1}{n}\sum_{i=1}^n l_{\theta}(X_i)
> $$
>
> where $l_{\theta}$ is a loss function that depend on $\theta$. Note that the empirical risk is a random variable since $X^n$ is a random variable. We can also take the expectation of the empirical risk $L_n(\theta)$ to get the expected risk:
>
> $$ E_{X^n} L_n(\theta) = E_X l_{\theta}(X)  $$
>
> When we write $\arg \min_{\theta} L_n(\theta)$, it is implicitly assumed that the n data samples are already drawn from its distribution, then $L_n(\theta)$ can be calculated deterministically. Now in our case, $L_{x,p}$ is a random variable. When we write $\arg \min_{\theta} L_{x,p}$, we implicitly assume the data samples
>
> {$ x^i, f_{I(x^i)}(x^i)$ } $_{i=1, \cdots, K}$
>
> are already drawn from its distribution, thus $L_{x,p}$ can be calculated deterministically.
>
>
>
>
> **Q: Is it difficult or tricky to tune the weight of the conditioning number loss term? What happens if the weight on this term is not high enough? What if it is too high? Ablation analyses of these factors would improve the paper**
>
> A: In Eq 7, the loss $L_{AE} = L_{rec} + \lambda_{cond} L_{cond}$.
> - If $\lambda_{cond}$ is too high, $L_{AE}$ will be dominated by $L_{cond}$.  Then the autoencoder may not reconstruct the microscopic dynamics well, hence may not capture the closure terms well. If the latent space is not closed enough, we can not learn the macroscopic dynamics well.
>
> - If $\lambda_{cond}$ is too low, $\varphi^{\prime}(x)$ may be very ill-conditioned. In Theorem 1, the eigenvalues of $\varphi^{\prime}(x)^T \varphi^{\prime}(x)$ is lower bounded by $b_1$ and upper bounded by $b_2$, and we have:
> $$
> b_1(L_x + C) \leq L_z \leq b_2 (L_x + C)
> $$This theoretically guarantees the effectiveness of $L_x$ when $b_1$ and $b_2$ are close. If $\varphi^{\prime}(x)$ is ill-conditioned, then $b_1$ is much smaller thant $b_2$. Theorem 1 can not guarantee the effectiveness of $L_x$ anymore.
>
> In our experiments, the value of $\lambda_{cond}$ is determined through a logarithmic grid search (more specifically, 1e-7, 1e-6, 1e-5, 1e-4, 1e-3). Thus in our experience the tuning $\lambda_{cond}$ is not very tricky.  Based on our experience, tuning $\lambda_{cond}$ is relatively straightforward and not tricky.
>
> Below we show the test error of the Predator-Prey system. We vary $\lambda_{cond}$ and train the autoencoder, then learn the macroscopic dynamics with $L_{x,p}(p=1/5)$ and report the test error.
> |  | | | | | | | |
> |  ----                                | ----      |----         | ----             |----        |----            |----                 |----             |
> |      $\lambda_{cond}$    | 0     | 1e-8  |    1e-7    |  1e-6 |   1e-5   |     1e-4     |    1e-2   |
> |  Test error of $L_{x,p} (p=1/5)$             | 6.42e-02 | 2.62e-03 | 2.84e-03 | *8.36e-04* | 3.66e-03 | 2.60e-03 | 4.24e-03|
>
> We can observe that the performance of $L_{x,p}$ may deteriorate when the parameter $\lambda_{cond}$ is either too high or too low.

---

> > ### Comment · Reviewer_RRQb · 2024-08-09
> >
> > I appreciate the authors' detailed response to my concerns and will raise the score to 7. However as I am not as familiar with existing literature on reduced order models as other reviewers, I will decrease my confidence to 3.

---

### Official Review · Reviewer_ktyB · 2024-07-11

**Soundness:** 3
**Presentation:** 3
**Contribution:** 3
**Rating:** 7
**Confidence:** 3

**Summary:**

The authors aims to use ML to compute macroscopic dynamics of a system from partially observed microscopic dynamics. The paper defines the macroscopic dynamics to be the lower dimensional latent space of an autoencoder that encodes the microscopic system. Given the autoencoder they train a macroscopic dynamics model using the partially observed forces of the microscopic dynamics that are projected to the macroscopic space via the encoder. Experiments are carried out on toy systems and lennard jones potential systems that are scaled to a large number of particles.

**Strengths:**

1. The paper tackles an impactful problem with important scientific applications
2. The underlying idea is simple (training an autoencoder to obtain the mapping from fine to coarse dynamics and using it to project the microscopic forces). However, non-trivial technical problems arise which the authors solve with creative techniques that would be valuable to share and present at a conference.
3. Rigorous evaluation: The authors have several creative insightful evaluations that go beyond the experiments that I would have imagined or expected (this is with the caveat that I do not work on reduced order modeling and am not familiar with what the standard evaluations are).

**Weaknesses:**

1.  Experiment system choices: The authors carry out experiments on small toy systems and large lennard jones systems of the same particles. It seems to me that the problem should become significantly harder with the "homogeneity" of the modeled system decreasing. It further seems to me that this would be the case in real scientific applications where we potentially have different atom types and multiple forces beyond lennard jones potential forces. Is this assessment correct? It would be great if you could put into perspective how close the experiments you carry out and the systems you choose are to actual scientific applications of interest. Thanks!
2. Presentation: you briefly mention closure modeling in the related work. Then in the methods section you say you follow closure modeling and define quantities such that a closed system arises. At this point I ask myself, what does closed mean? I hoped it becomes clear later in the paper but find there to be no sufficient explanation. Could you please let me know how you train the autoencoder in such a manner that the latent space captures specific desired macroscopic dynamics?
Minor:
1. Presentation: Motivation: What are some compelling application examples in which learning from partially observed microscopic dynamics is valuable?

**Questions:**

1. What if the particles to observe are not chosen uniformly at random? Is that also a common scenario where we do not have a uniform subsampling but our observations are biased by the specific subregion that we observe.

**Limitations:**

The paper prominently discusses two limitations (sparsity assumption and sampled distribution of microscopic dynamics) which are significant and put the paper into perspective of the field of work. I do not see further meaningful limitations that should be discussed apart from the possible limitation in terms of realistic evaluation which I mention in the weaknesses section.

---

> ### Author Rebuttal · Authors · 2024-08-07
>
> Thank you for your detailed review and thoughtful feedback. Below are our responses:
>
> **Q: What does closed mean?**
>
> A: Let the state of a system be $z$. When we say the system is closed, we mean the dynamics of the system can be written in the following form:
> $$
> \dot{z} = f(z)
> $$where $f$ is a function that only depends on the system's state $z$, not on any external variables. Thank you for pointing this out. We will include an explanation of this in the paper.
>
> **Q: Could you please let me know how you train the autoencoder in such a manner that the latent space captures specific desired macroscopic dynamics?**
>
> A: The autoencoder is trained with Eq. 7 in the paper:
> $$
> L_{AE} = L_{rec} + \lambda_{cond} L_{cond}
> $$ which is a reconstruction loss plus a condition number normalization loss.
>
> The minimization of the reconstruction loss ensures the latent space captures almost all the key dynamics and structures in the high-dimensional system. Then the specific macroscopic observables along with the latent space can be viewed as a closed system since the latent space captures almost all the information. The autoencoder only finds the closure to the macroscopic observables. Next, We use a neural network to parametrize their dynamics and train the model with $L_{x, p}$ to capture the desired macroscopic dynamics.
>
>
> **Q: The problem should become significantly harder with the "homogeneity" of the modeled system decreasing, In real scientific applications where we potentially have different atom types and multiple forces beyond Lennard-Jones potential forces. Is this assessment correct? It would be great if you could put into perspective how close the experiments you carry out and the systems you choose are to actual scientific applications of interest**
>
> A: Thank you for the insightful question. The application of our method to real scientific problems is also a primary focus of our next step. We agree that compared to the Lennard-Jones system we choose, we potentially have 1. different atom types and 2. more complex forces in real scientific applications.
>
> In the experiment of the Lennard-Jones system, we have validated that our method can be applied to particle systems and can scale to very large systems. We think the main challenge posed by different atom types and more complex forces lies in the closure modeling. More advanced autoencoders can be used for efficient closure modeling. Once the closure to the desired macroscopic observables is found, our framework can be readily applied to learn the macroscopic dynamics in real large-scale scientific problems.
>
>
>
> **Q: What are some compelling application examples in which learning from partially observed microscopic dynamics is valuable?**
>
> A: For example, in the design of Li-ion batteries, the viscosity and ionic diffusivity of liquid electrolytes are usually of key concern. But to obtain the macroscopic observables, existing method require long-term microscopic simulation where the calculation of the microscopic forces on all the atoms is extremely expensive [1]. In this situation, our method can learn the desired macroscopic dynamics from partial computation of microscopic forces, then we can significantly reduce the computational cost for force computations.
>
> **Q: What if the particles to observe are not chosen uniformly at random? Is that also a common scenario where we do not have a uniform subsampling but our observations are biased by the specific subregion that we observe**
>
> A:  When we do not have a uniform subsampling, we can use importance sampling to reweight the data such that the probability of each particle being chosen is the same. For example, assume a region $\Omega$ is divided into four subregions $\Omega_i, i=1,2,3,4$. The i-th region is chosen with probability $p(\Omega_i) = p_i >0$, and $p_1 + p_2 + p_3 + p_4 = 1$ . We have $n$ samples drawn from the distribution. Then we can reweight each sample from region $\Omega_i$ by $1/p_i$. In the reweighted data, each particle is chosen with the same probability.
>
> ---
> References:
>
> [1] Jia, W., Wang, H., Chen, M., Lu, D., Lin, L., Car, R., ... & Zhang, L. (2020, November). Pushing the limit of molecular dynamics with ab initio accuracy to 100 million atoms with machine learning. In SC20: International conference for high performance computing, networking, storage and analysis (pp. 1-14). IEEE.

---

> > ### Comment · Reviewer_ktyB · 2024-08-10
> >
> > Sorry for the late response - I reread the paper and will be faster to respond from now on.
> > (and thanks for the careful responses and explanations)
> >
> > I think your answers that you put under these headings best relate to my main concern:
> >
> > 1. "Q: Could you please let me know how you train the autoencoder in such a manner that the latent space captures specific desired macroscopic dynamics?"
> > 2. "Q: What are some compelling application examples in which learning from partially observed microscopic dynamics is valuable?"
> >
> > I think your answer to the first question misses my concern a little bit - I should have explained my question better. I understand that you train an autoencoder and a model operating on its latent space to capture the dynamics of the latent space.
> >
> > The concern is: why would the latent space capture the DESIRED macroscopic dynamics. If it is closed, it will capture all dynamics - sure. Also, it will be lower dimensional and capture SOME macroscopic dynamics. But why would the latent dimensions necessarily correspond to any of the macroscopic observables that "we" as the practitioners were interested in?
> >
> > For instance, considering your example that you gave for question 2 where we care about the viscosity of the system: what guarantees us that any of the latent variables correspond to viscosity? It seems to me that I am missing the entire point of the paper and that the latent representations remain uninterpretable and we cannot ensure that they correspond to certain macroscopic observables that we were interested in.

---

> > > ### Author Response · Authors · 2024-08-12
> > >
> > > Thanks for your insightful question. Let us address the question more clearly.
> > >
> > > In l 124, we mention '*we will use an autoencoder to find the closure $\hat{z} = \hat{\varphi}(x)$ to $z^{\ast}$ such that $z = (z^{\ast}, \hat{z})$ forms a closed system*'.
> > > Here $z^{\ast} = \varphi^{\ast}(x)$ is the macroscopic observables that we are interested in, and the dynamics of $z^{\ast}$ is the DESIRED macroscopic dynamics.
> > > The function $\varphi^{\ast}$ **is determined beforehand** and contains no trainable parameters. The closure $\hat{z} = \hat{\varphi}(x)$ is learned by the autoencoder. During the training of the autoencoder, $\varphi^{\ast}$ is fixed all the time, and only the parameters of $\hat{\varphi}$ are updated.
> > >
> > > For example, in the Lennard-Jones experiment, we choose the macroscopic observable $z^{\ast}$ as the Temperature $T$.  $z^{\ast} = \varphi^{\ast}(x)$ is defined through Eq 18:
> > > $$
> > > T = \frac{2}{3(N_{atoms}-1)} \times \sum_{i=1}^{N_{atoms}} \frac{m_iv_i^2}{2}
> > > $$
> > > In l 250, we mention we find another 31 closure variables using the autoencoder.
> > > We **directly concatenate** the macroscopic observable $z^{\ast}$ and the closure $\hat{z}$ to get the latent variable $z$:
> > > $$z = (z^{\ast}, \hat{z})$$
> > > Then the first dimension of $z$ is the desired macroscopic observable.
> > >
> > > The method mentioned above is common in closure modeling. For example,  the authors in [1] want to learn the stretching dynamics of polymer. Then they fix $z^{\ast}$ to be the length of the polymer and find another two-dimensional closure variable $\hat{z}$.  As for the interpretability, $z^{\ast}$ is interpretable since it is exactly the macroscopic observables we want. Usually, the closure variables $\hat{z}$ are not easy to interpret, since they are learned by the neural network. For simple systems like a nonlinear pendulum system, it may be possible to interpret $\hat{z}$ learned by the autoencoder [2, 3].
> > >
> > > We hope that this explanation clarifies your question.
> > >
> > >
> > > ---
> > > References:
> > >
> > > [1] Chen, X., Soh, B.W., Ooi, ZE. et al. Constructing custom thermodynamics using deep learning. Nat Comput Sci 4, 66–85 (2024). https://doi.org/10.1038/s43588-023-00581-5
> > >
> > >
> > > [2] Champion, K., Lusch, B., Kutz, J. N., Brunton, S. L. (2019). Data-driven discovery of coordinates and governing equations. Proceedings of the National Academy of Sciences, 116(45), 22445-22451.
> > >
> > > [3] Evangelou, N., Giovanis, D. G., Kevrekidis, G. A., Pavliotis, G. A., & Kevrekidis, I. G. (2023). Machine Learning for the identification of phase-transitions in interacting agent-based systems. arXiv preprint arXiv:2310.19039.

---

> > > > ### Comment · Reviewer_ktyB · 2024-08-12
> > > >
> > > > Thank you! This explanation fully addresses my fundamental concern of "how do you make sure that the learned dynamics correspond to the desired dynamics" and allows me to fully assess the paper. I am not sure whether I am the only reader who needs this more explicitly stated, and I leave it up to your assessment whether or not it would be beneficial to change the text.
> > > >
> > > > I am changing my overall score from 6 to 7 and my confidence has increased through the clarifications of the authors.

---

> > > > > ### Author Response · Authors · 2024-08-12
> > > > >
> > > > > Thank you for the suggestion! We will revise Section 4.1 to state this idea more explicitly and make it easier to understand.

---

### Official Review · Reviewer_d4vx · 2024-07-12

**Soundness:** 3
**Presentation:** 2
**Contribution:** 3
**Rating:** 5
**Confidence:** 4

**Summary:**

The authors describe a method to efficiently obtain aggregate information about forces acting on all particles in a system, in an effort to compute dynamics of macroscopic (aggregated) quantities. The key idea described in the paper is to sub-sample the particles to a small set, and only compute the forces on the small set (instead of computing all forces). The forces are then used to train a vector field of a latent space variable (through the chain rule). The latent space is obtained by training an auto-encoder on the microscopic states, independent of the force computations. The authors demonstrate the efficiency on multiple numerical systems, and include theoretical insights into the loss functions on micro- and macro (latent) states.

**Strengths:**

Obtaining dynamics of macroscopic variables from microscopic simulations is an important and challenging problem. Sub-sampling particles to reduce the burden on force computations and learning latent spaces of microscopic states automatically (instead of prescribing known quantities) are reasonable approaches to address this challenge. The chosen particle systems in the computational experiments are well-known and reasonable choices, and the reduction in the number of force computations (while keeping an accurate macroscopic state dynamics) are impressive.

**Weaknesses:**

1) The methods proposed in the paper are (a) sub-sampling particle indices to reduce number of force computations and (b) using a standard auto-encoder network to obtain a latent space. The former idea - especially when combined with the chain rule for the macroscopic dynamics - is noteworthy. Still, the general advancement of the state of the art is not large enough to warrant acceptance, in my opinion. In particular, the choice of auto-encoder is not suitable for the given problem (see question 6).

2) Some parts of the literature are not cited.

[A] l.65: reduced order modeling has not started in 2020 with deep learning methods. Appropriate literature from the decades before should be cited. A review can be found here:

[A1] Schilders, Wilhelmus H.A., Joost Rommes, and Henk A. van der Vorst, eds. Model Order Reduction: Theory, Research Aspects and Applications. Mathematics in Industry. Springer Berlin Heidelberg, 2008. https://doi.org/10.1007/978-3-540-78841-6.

[B] l.55: similarly, "learning from partial observations" is a decades (if not centuries) old problem. Key mathematical contributions are from Ruelle and Takens over 40 years ago, refined by Yorke:

[B1] Takens, Floris. “Detecting Strange Attractors in Turbulence.” Lecture Notes in Mathematics, 1981, 366–81. https://doi.org/10.1007/bfb0091924.
[B2] Ruelle, David, and Floris Takens. “On the Nature of Turbulence.” Commun. Math. Phys. 20, no. 3 (September 1971): 167–92. https://doi.org/10.1007/bf01646553.
[B3] Sauer, Tim, James A. Yorke, and Martin Casdagli. “Embedology.” Journal of Statistical Physics 65, no. 3 (1991): 579–616. https://doi.org/10.1007/BF01053745.

[C] In general, the idea of "sparse sampling" of the microscopic dynamics has also been published before. This must not just be cited but critically compared to, especially [C2].

[C1] Samaey, Giovanni, Ioannis G. Kevrekidis, and Dirk Roose. “Patch Dynamics with Buffers for Homogenization Problems.” Journal of Computational Physics 213, no. 1 (March 2006): 264–87. https://doi.org/10.1016/j.jcp.2005.08.010.
[C2] Liu, Ping, Giovanni Samaey, C. William Gear, and Ioannis G. Kevrekidis. “On the Acceleration of Spatially Distributed Agent-Based Computations: A Patch Dynamics Scheme.” Applied Numerical Mathematics 92 (June 2015): 54–69. https://doi.org/10.1016/j.apnum.2014.12.007.

[D] The idea of creating a latent space with an auto-encoder is of course also not new. The authors cite a few papers in l.143, but should do so in section 4.1.


3) l141: it is misleading to write "$g_\theta = \phi'(x)f(x)$". g depends on z, not x (correct?). It is not clear if the authors mean "$g_\theta(z) = \phi'(\phi^{-1}(z))f(\phi^{-1}(z))$, i.e., as a combination of $\phi'$ (encoder Jacobian), $\phi^{-1}$ (decoder) and $f$. At this point in the paper, it could also be that training of g happens directly on dynamics of z and avoid $\phi$. The latter is probably the case, looking at equation 9, but it is not clear from the text.


Minor:
1) l74: "where" instead of "here", and it is $f(x)\in\mathbb{R}^N$, not $f\in\mathbb{R}^N$ (the former is a vector, the latter a function).
2) l157: "we constrict the condition number..." (I think "constrain" is meant?) is not ideal wording; because the loss in eq. 6 does not constrain the condition number at all, it just penalizes it.

**Questions:**

1) What could be done with quantities that do not rely on individual molecule evaluations (e.g. force), but on integral quantities (e.g. "density")?
2) I assume the authors require all particles for the force computation because even if a particle is not chosen for force computaiton, it still needs to be present because other particles require its position for "their" force computation? This would also answer question 1.
3) What is the core difference between reference C2 and the approach by the authors?
4) l159: "we get rid of the matrix-vector product...": why is this beneficial? The new loss also has a matrix vector product with phi' and g.
5) Equation 2 is not precise enough (and contains two typos: the parenthesis and no period at the end). What is the "almost equivalent" here? What is assumed, precisely? How accurate does the approximate force need to be?
6) The autoencoder described in section 4.1 seems to have a very simple architecture. How is it possible to train it for such high-dimensional microscopic input states, if the encoder is not permutation invariant? For example, if the model is trained using a fixed index for each molecule, and then the indices are shuffled at random, the input to the autoencoder completely changes while the conformation (independent of the index) does not change at all. Since the autoencoder is not permutation invariant, it would need to learn all conformations including permutations, which is an incredibly high-dimensional space (not possible to sample in reasonable time).

**Limitations:**

The authors do not discuss the limitations of the auto-encoder they use to map to a latent space, in particular regarding permutation invariance of the molecular indices (see my question).
Societal concerns are not addressed at all; e.g. that speeding up force computations can speed up insights into materials that are harmful (not just beneficial).

---

> ### Author Rebuttal · Authors · 2024-08-07
>
> Thank you for your detailed review and thoughtful feedback. We have corrected all the minor issues and typos that you pointed out. We have also cited the previously missing literature you mentioned. Below are our detailed responses to the questions:
>
> **Q: Equation 2 is not precise enough. What is the "almost equivalent" here? What is assumed, precisely? How accurate does the approximate force need to be?**
>
> A: Thanks for pointing this out. In Eq 2, we mean $f_i(x_1, \cdots, x_n)$ can be calculated accurately through the microscopic coordinate of at most $M$ particles. The approximation is accurate enough such that the error is neglectable. For example, it is a common practice in molecular dynamics simulations to employ a cutoff distance for force fields. We have changed the '$\approx$' in Eq 2 to '$=$'.
>
> If we want to formalize Eq 2 more rigorously, we can require the error to be bounded by a tolerance $\epsilon$:
> $$
> ||f_i(x_1, \cdots, x_n) -  \tilde{f_i}|| < \epsilon
> $$ where $f_i$ is the exact microscopic force and $\tilde{f}_i$ is the approximation.$\epsilon$ will play a role in Theorem 1 such that Eq 11 becomes:
> $$
> b_1(L_x(\theta) + C) + \mathcal{O}(\epsilon) \leq L_z \leq b_2(L_x(\theta) + C) + \mathcal{O}(\epsilon)
> $$
> The performance of $L_x(\theta)$ can still be guaranteed if $\mathcal{O}(\epsilon)$ is much smaller compared to $b_1 C$.
>
> **Q: I assume the authors require all particles for the force computation because even if a particle is not chosen for force computations, it still needs to be present because other particles require its position for "their" force computation?**
>
> A: Not really. By the sparsity assumption mentioned in l 100, the microscopic force on each particle $i$ depends only on the microscopic coordinates of several particles that are in $J(x_i)$. If we want to calculate the forces on particles $x_1, \cdots, x_m$, we only need the coordinates of the particles that belong to $J(x_1), \cdots, J(x_m)$. The coordinates of particles in $J(x_i)$ are only used for the force calculation on particle $i$, not for any other particles. The computation cost increases linearly with the number of particles for which the microscopic forces are calculated.
>
> We also want to clarify that, since the macroscopic observable $z^{\ast}$ depends on the microscopic coordinate of all the particles, the calculation of $z^{\ast}$ needs all the particles to be present. But the calculation of $z^{\ast}$ is fast and cheap and the computation cost is almost neglectable.
>
> **Q: l159: "we get rid of the matrix-vector product...": why is this beneficial? The new loss also has a matrix-vector product with phi' and g.**
>
> A: The i-th entry of $\varphi^{\prime}(x)f(x)$ is $\sum_{i}\varphi_{ij}^{\prime}(x)f_j(x)$. It is difficult to find an unbiased estimation of $\sum_{i}\varphi_{ij}^{\prime}(x)f_j(x)$ using a subset of {$f_j(x)$}$_{j=1, \cdots, n}$.
>
>
> However, in the loss $L_x$ shown in Eq 10, $||f(x) - (\varPhi^{\prime}(x))^{\dagger}g_{\theta}(z)||_2^2$  can be written as
>
> $$\sum_{i=1}^n ||f_i(x) - ((\varphi^{\prime}(x))^{\dagger} g_{\theta}(z))_i||_2^2$$
>
> For any subset {$f_j(x)$}$_{j \in I(x)}, |I(x)| = n \cdot p$,
>
> $$ \frac{1}{p}\sum_{i\in I(x)}||f_i(x) - ((\varPhi^{\prime}(x))^{\dagger}g_{\theta}(z))_i||_2^2$$
>
> which can be used as a stochastic and unbiased estimation of $||f(x) - (\varPhi^{\prime}(x))^{\dagger}g_{\theta}(z)||_2^2$. The new loss can learn the macroscopic dynamics from a subset of microscopic forces and requires less force computation compared to $L_z$. The computation cost reduction is manifested through less force computations required by the new loss.
>
> **Q: What could be done with quantities that do not rely on individual molecule evaluations (e.g. force), but on integral quantities (e.g. "density")?**
>
> A: Our method can learn the dynamics of any macroscopic observables that can be written as a differential function of the microscopic coordinate: $z^{\ast} = \varphi^{\ast}(x)$. The density within a fixed region $\Omega$ is defined as the ratio of the number of particles contained in $\Omega$ to the area of $\Omega$:
> $$
> \frac{\sum_{i} \mathbb{1}(x_i\in \Omega)}{|\Omega|}
> $$The above function is not differentiable, but we can use a differential function to approximate it. Then our method can be applied.
> Currently, our method still can not address the case where $\varphi^{\ast}$ is not differentiable, or not deterministic.
>
> **Q: The autoencoder described in section 4.1 seems to have a very simple architecture. How is it possible to train it for such high-dimensional microscopic input states, if the encoder is not permutation invariant?**
>
> A: Thank you for the insightful question. Yes, we use MLP for both the encoder and the decoder, which indeed have a very simple architecture. In I 587, we mentioned each configuration has the same atom positions and velocity directions, and we only vary the temperature of the system. The order of the particles is fixed in the input to the autoencoder. In this way, no permutation needs to be considered and our simple MLP is enough to learn a latent space. We mainly want to use the experiment on the Lennard-Jones system to validate that our method can be applied to particle systems and can scale to very large systems. Note that in PDE experiments, the initial configuration is sampled from the initial distribution, and our method can handle such case.
>
> Thank you for pointing this out. We have added the limitation of the autoencoder in our paper. We will also try more advanced autoencoders such as graph neural networks that are permutation invariant to further improve our paper.
>
> **Q: What is the core difference between reference C2 and the approach by the authors?**
>
> A:  We answer this question in the top-level Author Rebuttal [C].

---

> > ### Comment · Reviewer_d4vx · 2024-08-10
> >
> > I thank the authors for their answers and clarification. The discussion of the literature is adequate. Still, my concern about major advancement of the field of networks (and ML in general) stands - enforced by the clarification that the auto-encoder indeed cannot deal with permutations of the individual atoms. I will keep the review score at 3.

---

> > > ### Comment · Reviewer_ktyB · 2024-08-11
> > > **A neat trick indeed! Why reject because permutation sensitive architecture in toy experiment?**
> > >
> > > I agree with d4vx that microscopic space loss is interesting and a neat trick (if I gathered that opinion correctly)!
> > >
> > > It may be a weakness of the paper (I think it is) that it only evaluates on rather toy tasks (one is quite a large scale but, as far as I understand, should still be considered toy because there is no heterogeneity).\
> > > However, in the large scale toy task, the permutation equivariance seems very much beside the point - generalization along this axis is not what we are concerned in evaluating. There is no reason to believe that a GNN wouldn't work if that generalization is required. \
> > > The paper's experiments are not meant to show that their implementation is optimal or an "advancement to networks" and addresses the concrete task (toy task) as well as possible, but rather provides evaluation and a sanity check of a neat new method. I think that can be considered interesting ML.

---

> > > > ### Comment · Reviewer_d4vx · 2024-08-11
> > > >
> > > > As I see it, the main idea of the paper is to utilize a small sample of all microscopic forces $f$ from a particle system to learn the vector field of macroscopic dynamics $g$ on learned latent variables $z$. The latter part is not performed with the right tools (GNNs could work, indeed). I agree that that is not directly related to the main idea, and changing to another architecture could make the examples more general. However, the main idea is not related to ML, but rather to molecular dynamics (MD). In fact, any reasonably efficient MD solver would probably already subsample forces to compute the microscopic dynamics, especially in the cases that the authors mention (e.g. forces that decay quickly with distance). I still fail to see how the main idea advances the state of the art in ML. For a conference on MD, this contribution may be more suitable.

---

> > > > > ### Author Response · Authors · 2024-08-12
> > > > >
> > > > > Thank you for the reply. We wish to clarify certain aspects of our paper.
> > > > > First, our paper is centered on machine learning for the following reasons.
> > > > >
> > > > > - The key contribution of our paper is the novel training loss $L_{x,p}$.
> > > > >     By training with $L_{x,p}$, we map the training procedure on the macroscopic coordinates back to the microscopic coordinates, on which partial force computations can be used as stochastic estimation to update model parameters. From this perspective, our method is directly related to ML,
> > > > >     as it resolves a key bottleneck in learning reduced macroscopic (closure) models for large systems.
> > > > > - Our method enables learning macroscopic dynamics from partial microscopic observations using ML.
> > > > >     In our paper, we validate our method in several experiments, not only including microscopic dynamics described by MD,
> > > > >     but also PDE based microscopic models.
> > > > >     Hence, our method applies not only to MD, but rather it addresses some common challenges in *AI for science*,
> > > > >     namely learning macroscopic dynamics from data.
> > > > >
> > > > > Next, we would like to clarify the distinction between our method and the sub-sampling techniques employed in the MD solvers you mentioned:
> > > > > - The sub-sampling method employed in MD solvers sub-samples a subset of the interactions between particles. For example, the random-batch molecular dynamics (RBMD) in [1] sub-samples the non-bonded interaction for each integration step. The force on all the atoms is still calculated. However, during the calculation of the force on each atom, only a subset of the interaction with other particles is considered. In our paper, when we refer to 'partial computation of microscopic forces', we only calculate the forces on a subset of particles.
> > > > >     In contrast, our method does not at any instant compute the forces on all particles, since this is not
> > > > >     necessary to obtain a (stochastic) approximation of the effective force on the macroscopic variables.
> > > > >
> > > > > - The sub-sampling method is applied to microscopic MD simulation. The computational cost scales with the system size $N$. Our method instead learns the macroscopic dynamics and simulates the dynamics in macroscale.
> > > > >
> > > > > ---
> > > > > References:
> > > > >
> > > > > [1] Gao, W., Zhao, T., Guo, Y., Liang, J., Liu, H., Luo, M., ... Xu, Z. (2024). RBMD: A molecular dynamics package enabling to simulate 10 million all-atom particles in a single graphics processing unit. arXiv preprint arXiv:2407.09315.

---

> > > > > > ### Comment · Reviewer_d4vx · 2024-08-14
> > > > > >
> > > > > > Thank you for pushing me on this. I had a very hard time understanding the core idea of your method, also because there are several issues with the terminology used in the manuscript. I strongly urge you to revise most of the descriptions and terminology in the paper based on my feedback. I will raise my score to 5, but not higher, because I cannot read the revised version and the submitted version has too many issues with terminology and missing explanations for a higher score. I will outline what I understand now, regarding the setting, the main idea, and give hints on the terminology below.
> > > > > >
> > > > > > ## Main setting
> > > > > > The proposed method is designed to approximate a vector field ${g}$ on macroscopic variables - including some of interest, and some to guarantee a closed model. The former are given, the latter are the output of a trained encoder network. All macroscopic variables $z$ considered are functions of microscopic variables $x$, approximated by an encoder network $\phi$ s.t. $\phi(x)=z$.
> > > > > >
> > > > > > ## Main idea
> > > > > > Many types of macroscopic variables can be *estimated* from a small subset $I$ of the microscopic variables $x$ (e.g., the mean of 1000 numbers can be estimated by randomly sub-sampling only 100 of them and dividing by 100 instead of 1000). This is at the heart of many coarse-graining approaches in MD. The core idea of the authors - as I understand it now - is that randomly choosing microscopic variables and computing their individual effect on the macroscopic variables (with reweighting based on the sample size) can also be done between the vector fields $f$ and $g$. Instead of computing all entries of $f(x)$, the authors only compute entries $f_{I(x)}(x)$, but then reweight them so that it can be used to estimate $g$. I understand this in terms of coarse-graining: the fraction $p\in(0,1)$ of sampled particles are "copied" 1/p times (by reweighting), so that the total influence is still as before. Using identical copies of a subset, instead of all particles, introduces an error. That error is reduced based on the "averaging effects" introduced by computing the macroscopic variables $z$. As some of these macroscopic variables are *learned*, the map $\phi$ can even be adjusted so that the error is reduced.
> > > > > >
> > > > > > ## More comments
> > > > > > I would have liked to connect the error reduction through averaging with the requirement on "small condition number" of $\phi'(x)\phi'(x)^T$, maybe the authors can do this.
> > > > > > The fact that *no microscopic simulation* is required at all makes this method extremely appealing for MD applications. As soon as reasonable configurations of particles can be obtained, and the vector field (e.g. including forces) can be evaluated on a small sample, it is possible to approximate the macroscopic states (the encoder) and their dynamics.
> > > > > >
> > > > > > ## Terminology
> > > > > > 1) The function $f$ in Equation 1 is not a force, but this term is used throughout the paper. It is a *vector field*, and if $x$ is a position, then it can be called a velocity. Only if $\ddot{x}=f(x)$ would $f$ be a force, but then all derivations lateron for $\dot{z}$ etc. would need to be adjusted. It is clear why in most settings used in the paper, the complication to compute the vector field is in the computation of the forces, but the notation is misleading and must be cleared up.
> > > > > >
> > > > > > 2) The fact that you call it a "solver" in many places, while what you actually mean is a subroutine that computes forces - and does not "solve" anything - was the main reason I did not understand the appeal of your method. In my opinion, a solver "solves" something, usually a differential equation, so that the output is a solution (e.g. a trajectory). In the paper, the "black-box solvers" in figure 1 and algorithm 1 are actually not solving the microscopic dynamics, they are just computing the "forces" (or, rather, "evaluate the vector field"). This is the main reason I now think the method is great (as I discuss above), but it was extremely difficult for me to understand because of the mismatch of terminology (solver vs. vector field evaluation).
> > > > > >
> > > > > > 3) Similar to 2, Figure 1 also did not help to mitigate my main concerns. On the left there are many black arrows indicating the vector field (or "force") on each particle. But your method does not even need any of this in the "data set", correct? The figure suggests that the "black-box solver" must compute all the black arrows, and then only a subset of them is used in training. This is how an actual solver for the microscopic dynamics would do it, too - because the microscopic state must be advanced over time, and an actual solver can only do that after computing all forces for all particles (maybe subsampling which neighbors are used in the process). For your method, none of that is relevant, which I now understand, because you do not even need any trajectories on the microscopic scale for training the macroscopic vector field.

---

> > > > > > > ### Author Response · Authors · 2024-08-14
> > > > > > >
> > > > > > > Thank you very much for your valuable suggestions that helped us to improve our paper.
> > > > > > >
> > > > > > > We will revise our descriptions and terminology according to your suggestion to make our paper easier to understand:
> > > > > > >
> > > > > > > -  For the 'more comments part', given the same number of data points, the larger the $p$ is, the more force computation we will have, and theoretically the error will be smaller. Here the error is controlled through the force computation.Small condition number of $\varphi^{\prime}(x) \varphi^{\prime}(x)^T$ controls the error through the encoder.  We will try to explain this in more detail in the paper.
> > > > > > > -  We agree that calling $f$ as a vector field is more accurate. We will show the reason why we call $f$ the 'force' in the paper.
> > > > > > > As shown in Eq 3 and Eq 4, the Newton's law of motion in molecular system is :
> > > > > > > $$\dot{r_i} = v_i$$
> > > > > > > $$\dot{v_i} = \frac{1}{m_i}F_i$$
> > > > > > > here $f_i$ is the position, $v_i$ is the velocity, $m_i$ is the mass and $F_i$ is the force on the $i-th $ atom. In this case the microscopic state $x = (r_i, v_i)$, and its dynamics $f = (v_i, \frac{1}{m_i} F_i)$. $\frac{1}{m_i} F_i$ is part of the $f$ in molecular dynamics, then we call $f$ 'force' in other cases such as the PDE case even though $f$ is not a 'force'. Thank you for pointing out that calling $f$ 'force' is difficult to understand. For notation simplicity, how about adding an explanation for the reason why we call $f$ the 'force' after Eq 1 in the paper as the following?
> > > > > > >
> > > > > > > *Note that the $f(x(t))$ is indeed a vector field. In molecular dynamics, ... (The above-mentioned molecular dynamics example)...for notation simplicity, we will call $f$ the force in the rest of the paper.*
> > > > > > >
> > > > > > > - You are correct that the 'black-box solver' only computes the 'forces', rather than the trajectory. We will modify the terminology to 'microscopic force calculator' or 'vector field calculator'. Our method does not need the trajectories of the microscopic system, but we assume we know a distribution of the microscopic state $x$. If the distribution is unknown, there are still several possible ways for our method to work, such as 1. using active learning to add the most informative configuration and 2. using cheaper coarse-grain or Markov chain Monte Carlo (MCMC) methods to sample an approximate distribution.
> > > > > > >
> > > > > > > - We will change Figure 1 to make it easier to understand. In the Data generation part of Figure 1, we will remove all the black arrows on the left of the 'black-box solver', and only preserve the pink arrows on the right of the 'black-box solver'.

---

### Author Rebuttal · Authors · 2024-08-07

We would like to thank all reviewers for providing detailed reviews and constructive feedback that have improved the paper.

Reviewer d4vx mentioned some relevant literature. We would like to compare these studies with our research to provide deeper insights into our method.

**[A]** A1 includes many model order reduction methods such as the Krylov Projection Framework and proper orthogonal decomposition. A1 also mentions the data-driven reduced order method but considers the dimension reduction mapping to be the linear combination of several chosen basis functions. In this work, we use the neural network to parametrize the dimension reduction mapping, which has more flexible form but harder optimization problem. We want to mention that our method can be generally combined with any dimension reduction method as long as they can learn the closure to the desired macroscopic observables well.



**[B]** In B1, B2, and B3, the 'partial observation' refers to an observable of the microscopic state $x$ (e.g. the observable is a function of $x$ ). These studies aim to obtain information (e.g., attractor) about the microscopic system from these macroscopic observables.

In our work, 'partial observation' refers to the computation of microscopic forces on a subset of particles. Our work aims to learn the macroscopic dynamics from microscopic forces.


**[C]** The approach in C1, C2 belongs to the equation-free framework in [1]. The core difference between these studies and our approach is:
- C1 and C2 only consider partial differential equation (PDE) systems. The macroscopic observables in C1 and C2 are the solution of the PDE at the coarse spatial grid.  Then the macroscopic observables depend locally on the microscopic coordinates. In our work, the macroscopic observables depend globally on the microscopic coordinates of all the particles. In the patch dynamics scheme mentioned in C1 and C2,  the lifting and restriction operator can be viewed as the encoder and decoder in our case. The lifting and restriction operator is explicitly defined, but we need to train the autoencoder, which may cause some optimization problems.
- The approach in C1 and C2 does not learn the macroscopic dynamics. Our method explicitly learns the macroscopic dynamics parametrized by a neural network.  When the closure to the macroscopic observables is difficult to learn, our method may not handle this case. However, the equation-free framework can still work by bypassing the derivation of the macroscopic evolution equations.
- In C1 and C2, during the simulation of macroscopic dynamics, microscopic simulation still needs to be performed in small spatial domains and for short times. In our method, once the neural network for parametrizing the macroscopic dynamics is trained, we can simulate the macroscopic dynamics directly in macroscopic space, without performing any microscopic simulation.

---
References:

[1] Gear, C. W., Hyman, J. M., Kevrekidid, P. G., Kevrekidis, I. G., Runborg, O., & Theodoropoulos, C. (2003). Equation-free, coarse-grained multiscale computation: Enabling mocroscopic simulators to perform system-level analysis.

---

### Decision · Program_Chairs · 2024-09-25

**Decision:**

Accept (poster)

**Comment:**

The reviewers find this work studying an important problem and encourage the authors to incorporate the suggestions and developments during the review process.